



# Age Spectra and Other Transport Diagnostics in the North American Monsoon UTLS from SEAC⁴RS In Situ Trace Gas Measurements

Eric A. Ray[1,2], Elliot L. Atlas[3], Sue Schauffler[4], Sofia Chelpon[5], Laura Pan[4], Harald Bönisch[6] and Karen H. Rosenlof[1]

[1]NOAA Chemical Sciences Laboratory, Boulder, CO, USA
[2]Cooperative Institute for Research in Environmental Science (CIRES), University of Colorado, Boulder, CO, USA
[3]University of Miami, Miami, FL, USA
[4]National Center for Atmospheric Research, Boulder, CO, USA
[5]City University of New York Graduate Center, New York, NY, USA
[6]Karlsruhe Institute of Technology, Institute of Meteorology and Climate Research, Karlsruhe, Germany

*Correspondence to*: Eric Ray (eric.ray@noaa.gov)

**Abstract**. The upper troposphere and lower stratosphere (UTLS) during the summer monsoon season over North America (NAM) is influenced by the transport of air from a variety of source regions over a wide range of time scales (hours to years). Age spectra are useful for characterizing the transport into such a region and in this study we use and build on recently developed techniques to infer age spectra from trace gas measurements with photochemical lifetimes from days to centuries. We show that the measurements taken by the Whole Air Sampler instrument during the SEAC⁴RS campaign can be used to not only derive age spectra, but also path-integrated lifetimes of each of the trace gases and surface source regions. The method used here can also clearly identify and adjust for measurement outliers that were influenced by polluted surface source regions. The results are generally consistent with expected transport features of the NAM but also provide a range of transport diagnostics that have not previously been computed solely from in situ measurements. These methods may be applied to many other existing in situ datasets and the transport diagnostics can be compared with chemistry-climate model transport in the UTLS.

## 1 Introduction

The UTLS in the region of the North American monsoon (NAM) is influenced by rapid transport from convection that penetrates from below, mixing due to wave breaking and transport in the lower branch of the Brewer-Dobson circulation, as well as slowly descending air from the stratosphere above (Weinstock et al., 2007; Boenisch et al., 2009; Orbe et al., 2015; von Hobe et al., 2021). The time scales of these different transport pathways from the surface to the UTLS range from hours to years and thus this part of the atmosphere, especially over the monsoon regions during summer, contains a uniquely complex dynamical history and chemical composition. A number of previous studies have used *in situ* trace gas measurements to estimate average transport characteristics of the summer UTLS such as the tropospheric fraction (Ray et al., 1999; Weinstock et al., 2007) and mean age of air (Boenisch et al., 2009; Birner et al., 2020). However, average transport quantities do not fully capture the complexity of the region and so more sophisticated transport descriptions, such as age spectra and surface source region identification, have also been estimated primarily from model simulations (Diallo et al., 2012; Orbe et al., 2015; Ploeger and Birner, 2016; Hauck et al., 2019). The modeled transport quantities provide useful information but are also dependent on a variety of input meteorological products that can produce different results (Ploeger et al., 2019).

Most previous studies of age of air from trace gas measurements have focused on the stratosphere and used species with very long lifetimes that are increasing in time, such as $CO_2$ and $SF_6$ (e.g. Andrews et al., 1999, 2001; Waugh and Hall, 2002). For these trace gases the age of air can be inferred from concentration differences between measurement locations and a source region, typically the tropical tropopause since this is where most of the air enters the stratospheric overworld. Trace gases with significant local photochemical loss along pathways from the source region to the measurement locations can also be used to calculate age of air, but in this case the path-integrated lifetimes need to be estimated to account for the photochemical loss. The path-integrated lifetime is different from the global and local photochemical lifetimes, and is unique to each measurement time, location and trace gas. This is complementary to the age of air since it reveals pathway information through known local photochemical loss regions (e.g. Hall, 2000; Moore et al., 2014). Schoeberl et al. (2005) was the first to estimate stratospheric age spectra using multiple trace gases with relatively short local stratospheric lifetimes, such as $N_2O$ and CFC-11, and to solve for the path-integrated lifetime of each trace gas at individual locations.

The UTLS presents more challenges to calculate age of air, path-integrated lifetimes and other transport diagnostics due to the wide range of possible source regions and relatively rapid transport and mixing compared to the stratospheric overworld. Recent studies have taken advantage of the predominance of emissions of anthropogenically produced trace gases in the Northern Hemisphere (NH) compared to the Southern Hemisphere (SH) to calculate interhemispheric transport diagnostics in the troposphere based on the differences in NH-SH surface trace gas measurements (Waugh et al., 2013; Holzer and Waugh, 2015).





The interhemispheric transport time scale is on the order of a year but to more generally calculate transport diagnostics at any location in the free troposphere from the surface requires resolving transport time scales of hours in the case of convective transport. This rapid transport can only be clearly detected in very short-lived trace gases (path-integrated lifetimes of days). Luo et al. (2018) used *in situ* trace gas measurements over the tropical Pacific Ocean with a range of lifetimes from days to centuries to derive the 'transit time distribution' from the surface to the tropical upper troposphere. The terminology 'transit time distribution' is commonly used in studies focused on the troposphere and is equivalent in meaning to the 'age spectrum' typically
used in stratospheric studies. The Luo et al. (2018) study was one of the first to bring stratospheric age of air techniques to tropospheric measurements and was unique in the use of such a wide range of trace gases with different lifetimes.

In the UTLS, age spectra derived from trace gas measurements have been somewhat rare (Ehhalt et al., 2007; Boenisch et al., 2009), although recent studies have developed new techniques with promising results (Luo et al., 2018; Hauck et al., 2019, 2020;
Podglajen and Ploeger, 2019). The studies of Hauck et al. (2019, 2020) focused on the lowermost stratosphere and an inverse technique following Schoeberl et al. (2005) with the tropopause as the source region. These studies introduce an imposed seasonal cycle on the age spectra based on model output as well as tropical and extratropical source regions following the known transport pathways to the lowermost stratosphere (e.g. Ray et al., 1999; Boenisch et al., 2009). The partitioning between the tropical and extratropical source regions is also prescribed based on model output. These studies are essentially a hybrid of
theoretical, observational and model analysis. The path-integrated lifetimes of the trace gases are estimated in Hauck et al. (2020) along with the age spectra but the lifetimes are not shown.

As a starting point for this work, we use the technique put forth in Luo et al. (2018). In addition to the use of a wide range of trace gases, this study also introduced the novel path-integrated lifetime vs. normalized mixing ratio framework, which we use
extensively. In the current study we use measurements taken during the Studies of Emissions and Atmospheric Composition, Clouds and Climate Coupling by Regional Surveys (SEAC[4]RS) mission in the UTLS over North America during the NAM, primarily from the Whole Air Sampler (WAS) instrument on the ER-2 aircraft. The fundamental differences in the calculation presented here compared to Luo et al. (2018) and to other previous studies are that, in addition to finding age spectra, we solve for surface source regions of air from the tropics and NH extratropics, and we find path-integrated lifetimes for each trace gas for
the sampled UTLS. We use a combination of measurements of 20 trace gases with a variety of photochemical lifetimes as well as $CO_2$, which together provide powerful constraints on the derived transport quantities.

We describe the data used in this work in Section 2 and the general method in Section 3. Results for the average theta profiles over the whole SEAC[4]RS mission and for individual measurement locations are shown in Section 4. The calculation performed
here has many details and assumptions, some of which are described more fully in the supplement.

**2 Data**

The UTLS data used in this study was taken during the SEAC[4]RS mission which took place in August and September 2013 over
North America (Toon et al., 2016). We used measurements of 20 different species from the Whole Air Sampler (WAS) instrument on the ER-2 (Table 1) as well as $CO_2$ measurements from the Harvard Picarro Cavity Ringdown Spectrometer (PCRS), $O_3$ from the NOAA Unmanned Aircraft Systems (UAS) $O_3$ instrument and water vapor from JPL Laser Hygrometer (JLH). The 20 species measured by WAS were chosen due to their range of lifetimes and lack of significant production in the atmosphere. We used the WAS merge data files (https://www-air.larc.nasa.gov/cgi-
bin/ArcView/seac4rs?MERGE=1#WAS.ER2_MRG/), which put all of the trace gas and meteorological measurements on the WAS sampling frequency of ~4-5 minutes. In total we used data from the 548 WAS sampling times during the mission above the 340 K potential temperature surface, although not all of the trace gases are available at all of the sampling times.

Surface measurements come from the NOAA Global Monitoring Laboratory (GML) network for 16 of the 20 trace gases and
from the Advanced Global Atmospheric Gases Experiment (AGAGE) network for three of the trace gases. 1,2-Dichloroethane is not regularly measured by either network so we used a constant mixing ratio in each latitude region based on previously published measurements (Class and Ballschmiter, 1987). $CO_2$ mixing ratios were obtained from NOAA's CarbonTracker, version CT2019B (Jacobsen et al., 2020).

*Table 1. List of trace gases used in this study, their estimated upper tropospheric lifetimes and the sources of the surface time series.*

| Trace Gas | Initial UT Lifetime (days)[1] | Surface Data Source | 30 Year Extension Source |
|---|---|---|---|
| n-Butane, $C_4H_{10}$ | 8 | NOAA GML stations[2] | Seasonal cycle |
| Isobutane, $C_4H_{10}$ | 9 | NOAA GML stations | Seasonal cycle |
| Propane, $C_3H_8$ | 14 | NOAA GML zm[3] | Seasonal cycle |
| Ethyne, $C_2H_2$ | 16 | NOAA GML zm | Seasonal cycle |





| 1,2-Dichloroethane, $C_2H_4Cl_2$ | 30 | Constant[4] | Constant |
|---|---|---|---|
| Ethane, $C_2H_6$ | 80 | NOAA GML zm | Trend[6] |
| Perchloroethylene, $C_2Cl_4$ | 100 | AGAGE stations[5] | Measured GR[7] |
| Chloroform, $CHCl_3$ | 200 | AGAGE stations | Measured GR |
| HFC-152a, $C_2H_4F_2$ | 600 | NOAA GML zm | Measured GR |
| Methyl Bromide, $CH_3Cl$ | 800 | NOAA GML zm | Measured GR |
| HCFC-141b, $CH_3CCl_2F$ | 3,000 | NOAA GML zm | Measured GR |
| HCFC-22, $CHClF_2$ | 4,000 | NOAA GML zm | Measured GR |
| HFC-134a, $CH_2FCF_2$ | 5,000 | NOAA GML zm | Measured GR |
| Halon-1211, $CBrClF_2$ | 6,000 | NOAA GML zm | Measured GR |
| HCFC-142b, $CH_3CClF_2$ | 7,000 | NOAA GML zm | Measured GR |
| Carbon Tetrachloride, $CCl_4$ | 10,000 | NOAA GML zm | Measured GR |
| CFC-11, $CCl_3F$ | 15,000 | NOAA GML zm | Measured |
| CFC-113, $C_2Cl_3F_3$ | 20,000 | NOAA GML zm | Measured GR |
| CFC-12, $CCl_2F_2$ | 25,000 | NOAA GML zm | Measured |
| CFC-114, $C_2Cl_2F_4$ | 60,000 | AGAGE stations | Measured GR |

[1]UT lifetimes refer to path-integrated lifetimes based on tropospheric and lower stratospheric local lifetimes of each trace gas from WMO (2018), Luo et al. (2018), Chelpon et al. (2021), Tang et al. (2007), Hodnebrog et al. (2018), Wuebbles et al. (2007), Olaguer (2002), Chipperfield et al. (2013) and Singh et al. (1996).

[2]NOAA GML stations used were Samoa, Mauna Loa, Key Biscayne and Wisconsin (LEF).

[3]NOAA GML zonal mean surface time series are calculated in seven different latitude bands from all available flask and in situ measurement sites. (https://www.esrl.noaa.gov/gmd/hats/data.html) (G. Dutton, personal communication).

[4]1,2-Dichloroethane time series is a constant in each latitude bin with latitude gradients based on the measurements of Class and Ballschmiter (1987).

[5]AGAGE stations used were Samoa, Ragged Point Barbados, Trinidad Head and Mace Head.

[6]Ethane growth rate from Helmig et al. (2016).

[7]Measured GR refers to the use of the growth rate based on the earliest available two years of measurements applied to extend the time series back to 1983.

The surface measurements are convolved with the UTLS age spectra to derive integrated surface boundary conditions for each trace gas. This calculation requires time series of each trace gas in a range of latitudes that span the expected source regions of air to the NH UTLS (Ray et al., 2004; Herman et al., 2017), roughly from the equator to the high northern latitudes (Figure 3). We extend the surface time series back 30 years prior to the time of the SEAC[4]RS mission. Beyond 30 years, the spectra frequencies are at least several orders of magnitude lower than the peak (modal) frequencies so the inclusion of older spectra

times does not significantly change the derived surface boundary conditions.

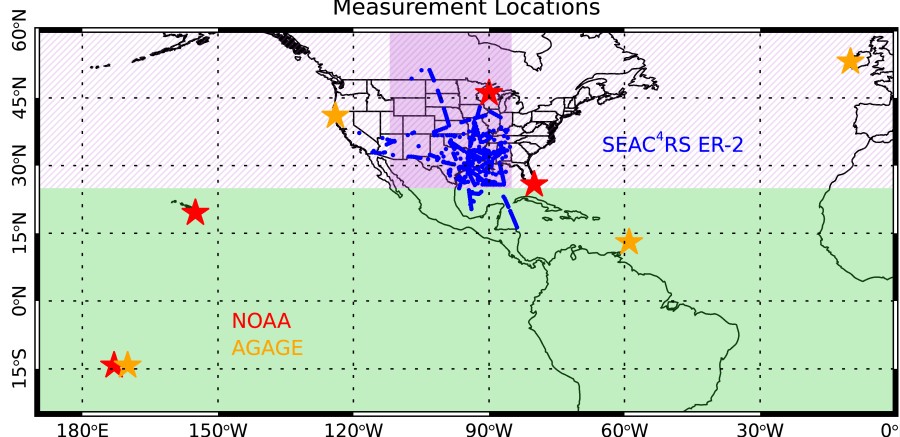

*Figure 1. Map of the location of SEAC[4]RS ER-2 in situ measurements in the UTLS (blue symbols) and surface NOAA (red) and AGAGE (orange) measurement stations used in the analysis. The darker shaded purple region is where CarbonTracker NAM*

*(1°x1° resolution) data was used to calculate NA $CO_2$ surface time series, and the light shaded green region indicates where CarbonTracker Global (3°x2° resolution) was used. For the other trace gases used in the analysis either individual surface stations (stars) or zonal means (light purple and green shaded regions) were used to calculate the tropical and NA time series (see Table 1).*





To construct each trace gas time series, we used surface network measurements when available (**Table 1**) and when measurements were not available, we extended the time series backward to the year 1983 based on published trends and calculated seasonal cycles. For the shortest-lived trace gases, such as propane, only the previous several months are important for deriving the surface boundary conditions so the seasonal cycle is sufficient to create the surface time series. For the longest-lived trace gases, such as the CFCs and HCFCs, we interpolated backward from the earliest available measurements using the

growth rate over the two subsequent years. Examples of these surface time series for ethane are shown in Figure 2. Both the seasonal cycle and latitudinal gradient are substantial for ethane and many other trace gases used in this study so the BL mixing ratios are sensitive to the surface source region as well as the age spectra.

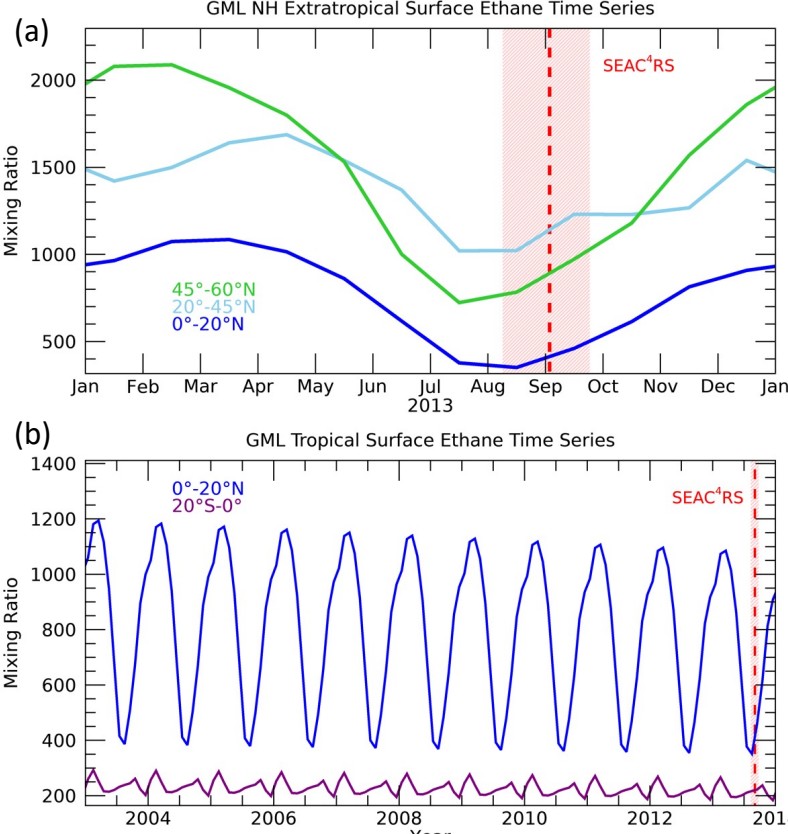


Figure 2. *Time series of ethane at the surface in (a) NH extratropical and (b) tropical latitude bands. The timing of the SEAC⁴RS mission is indicated by the red shading covering all of the flights and the dashed red line indicates the mission average date.*

The calculation of transport diagnostics also requires an estimate of the lifetime of each trace gas as shown in Table 1. The lifetimes in the calculation are path-integrated lifetimes from the surface source regions. There can be many pathways to a location in the UTLS so exact lifetimes are not known. The values shown in Table 1 are initial estimates of the path-integrated lifetimes of each trace gas in the sampled upper troposphere, specifically in the 350-360 K layer, based on local lifetime estimates in the troposphere and lower stratosphere. The sources of local lifetime estimates are listed in Table 1, with the

primary sources of tropospheric lifetimes from the Luo et al. (2018) and Chelpon et al. (2021) studies, and the primary source of stratospheric lifetimes from WMO (2018). Since the 350-360 K layer sampled during SEAC⁴RS was in the subtropical upper troposphere, the path-integrated lifetimes shown in Table 1 are weighted towards UT local lifetimes such as in Chelpon et al. (2021). These lifetimes are used as initial estimates in the calculation but are adjusted for each theta layer and each individual measurement location based on the best fit age spectra as described below. Thus, while it is important to have the initial path-



integrated lifetime estimates for the calculation, the final derived lifetimes are considerably different from those shown in Table 1.

CO$_2$ is used as an additional constraint on the age spectra calculated from the 20 trace gases listed in **Table 1** and thus 30 year surface time series over a range of latitudes are also necessary to convolve with the age spectra. CarbonTracker contains both a
global gridded CO$_2$ product as well as a higher resolution gridded product over North America. For surface latitudes in the region sampled by SEAC$^4$RS (20°-50°N) (Figures 1, S2) we use the North American (NA) product zonally averaged from 85°-112°W and for tropical latitudes we use full zonal means of the global product. The reason for this different range of surface longitudes is that it is assumed the influence from tropical latitudes will have traveled some distance and be relatively well mixed in longitude compared to the local convective influence in the sampled NA region.

For some of the trace gases a scaling was applied to account for calibration offsets between WAS measurements and those from the surface networks. These scaling values were estimated based on the best agreement with DC-8 WAS measurements near the surface during SEAC$^4$RS as well as expected relationships between normalized trace gases in the upper troposphere from previously calculated tropospheric lifetimes. The specific scaling values and methodology are provided in Section S1 of the
supplement. It should be noted that the scaling does not significantly affect the age spectra calculation and that the scaling factors are on the order of 1-5%.

### 3 Methods

The concentration of a trace gas, $i$, at a particular location $\boldsymbol{x}$ and time $t$ in the atmosphere can be expressed as

$$\chi_i(\boldsymbol{x},t) = \int_0^\infty \chi_i(\boldsymbol{x}_o, t-t') e^{-t'/\tau_i(x,t')} G(\boldsymbol{x},t,t') dt' \qquad (1)$$

where $t'$ is the transit time or age from a source region, $\tau_i$ is the path-integrated lifetime of the trace gas, $\chi_i(\boldsymbol{x}_o, t-t')$ is the
concentration time series at the source region $\boldsymbol{x}_o$ and $G$ is the age spectrum of all the paths from the source region to the location $\boldsymbol{x}$ (e.g. Schoeberl et al., 2000; Ehhalt et al., 2007; Hauck et al., 2019). The age spectra can be assumed to have a functional form following many previous studies of age of air in the stratosphere (e.g. Hall and Plumb, 1994; Schoeberl et al., 2005)

$$G(\boldsymbol{x},t,t') = \frac{z}{2\sqrt{\pi K(x,t)t'^3}} exp\left(\frac{z}{2H} - \frac{K(x,t)t'}{4H^2} - \frac{z}{4K(x,t)t'}\right) \qquad (2)$$

where $H$ is the scale height, $z$ is the altitude at location $\boldsymbol{x}$ and $K$ is an effective diffusion coefficient that varies with location and time. This functional form allows the age spectra to be defined by a single parameter $K$ at a particular altitude $z$. Thus, in the rest of the paper we refer to the age spectra with dependence on $K$, $z$ and $t'$.

In Luo et al. (2018) and Chelpon et al. (2021) it was shown that there exists a compact relationship between measured trace gas mixing ratios in the tropical upper troposphere (UT) normalized by local marine boundary layer (BL) mixing ratios, referred to as the measured UT fraction ($\mu^*$), vs. lifetime ($\tau$) over a wide range of trace gases. These studies made the approximations of steady state and 1D conditions in order to rearrange Equation 1 to solve for an idealized form of $\mu$

$$\mu_i(K,z) = \frac{\chi_i(z)}{\chi_{io}} = \int_0^\infty e^{-t'/\tau_i} G(K,z,t') dt' \qquad (3)$$

where $\chi_{io}$ is a constant BL mixing ratio of trace gas $i$ and $\chi_i(z)$ is an idealized UT mixing ratio of trace gas $i$ at height $z$. With this formulation, each age spectra $G$ produces a unique idealized $\mu - \tau$ relationship that can be compared to the measurement based $\mu^* - \tau$ relationship. The age spectrum that produces the best comparison to the $\mu^* - \tau$ relationship is assumed to be the
best approximation of the age spectrum at the measurement location and time.

In this study we do not assume steady-state or 1D conditions but we do utilize the $\mu - \tau$ and $\mu^* - \tau$ relationships to identify transport parameters that best describe the measurements. From Equation 1, we see that each trace gas time series at the source region $\chi_i(\boldsymbol{x}_o, t-t')$, which we take to be the surface BL, can vary with location $\boldsymbol{x}_o$, which we simplify to only consider the
latitudinal dimension, and time in the form of the age $t'$ compared to the UTLS measurement time $t$. These dependencies mean we need to construct BL time series for all of the trace gases at a range of latitudes as described in the previous section, but also that the normalized mixing ratios $\mu$ and $\mu^*$, which we will refer to as BL fractions, will have latitudinal source region and age dependencies.

The primary BL source latitudes to the sampled UTLS (Figures 1 and 3) are not known *a priori* so this is a free parameter in the calculation. Based on many previous studies of UTLS transport, we can apply some general constraints to the possible BL source latitudes and allow the calculation to optimize among them. The UTLS up to 400 K was shown in previous studies to





have been significantly influenced by convective transport from the North American continent during the SEAC[4]RS mission (e. g. Herman et al., 2017). Convective transport from the BL can occur on the time scale of hours, and if it enters the stratosphere

can remain there for days to months depending on the depth above the tropopause. The onset of NAM convection each year generally occurs in early June (e.g. Clapp et al., 2020), roughly two to three months before the August-September time frame of the SEAC[4]RS mission. Thus, the oldest air that would have been convectively transported from the local, extratropical surface to the sampled UTLS would likely be two to three months. This would represent air that was convectively injected in June or July into the lowermost stratosphere where it then slowly descended to the sampled location by August or September. There could

also be air from the NA surface more than 3 months old in the LS that was not convectively injected, but rather took a longer path via the tropical UT where it could then have been isentropically transported into the LS.

Air from the tropical surface can be transported to the extratropical UTLS initially by convection to the tropical transition layer and then either by isentropic transport or through the stratosphere by the Brewer-Dobson circulation (e.g. Boenisch et al., 2009).

The transport times for these pathways from the tropical surface range from days to decades (e.g. Ploeger et al., 2019) (Figure 3).

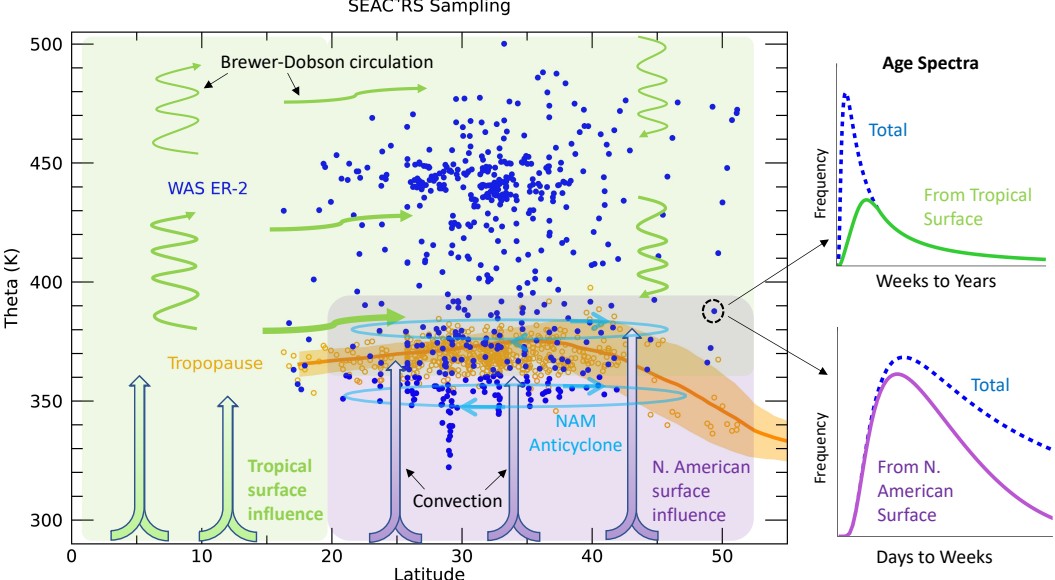

*Figure 3. Sampling of the WAS measurements on the ER-2 (blue symbols) during SEAC[4]RS as well as indicators of the*

*tropopause levels based on Microwave Temperature Profiler (MTP) measurements (orange symbols) and from MERRA2 averaged over the mission time period and sampling area (average in orange line and standard deviation in orange shading). The purple shading and thick arrows indicate convective transport from the North American surface to the UTLS, while the green arrows and shading indicate transport from the tropical surface to the sampled UTLS through convection followed by isentropic mixing and/or advection by the Brewer-Dobson circulation. The NAM anticyclone circulation is indicated by the light*

*blue ovals. Age spectra examples for an individual measurement location are shown on the right. The total spectrum is shown by the blue dashed lines in each plot and the partitioning of the spectrum into that from the tropical surface (light green, top) for longer time scales and from the North American surface (purple, bottom) for shorter time scales.*

Based on these transport characteristics of air from the surface to the extratropical UTLS we partition the age spectra into

transport from tropical and NA surface source regions. We assume the shortest time scales of the age spectra, hours to weeks, are primarily from NA latitudes and that longer time scales, weeks to years, are from tropical surface latitudes (Figures 1 and 3). This partitioning follows from previous modeling work (e.g. Orbe et al., 2013, 2015) that show a mixture of surface source region origins in the NH summer UTLS. The age spectra can be partitioned as

$$G_{TR}(K, z, t') = f(z, t')G(K, z, t') \tag{4}$$

$$G_{NA}(K, z, t') = (1 - f(z, t'))G(K, z, t') \tag{5}$$

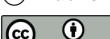



where $G_{TR}$ and $G_{NA}$ are the age spectra from tropical surface latitudes (25°S-25°N) and NA surface latitudes (25°-55°N), respectively. Note that $G_{TR}$ and $G_{NA}$ are not renormalized so only the total spectrum $G$ sums to one. The scaling factor $f$ has an age dependence with a Gaussian form such that $f(0\ \text{days}) = 0$ and $f(t_f'\ \text{days}) = 1$ (Figure S3). Thus, at $t_f'/2$ days the total age spectrum is evenly divided between tropical and NA surface source regions. The scaling factor is not known *a priori* so we ran the calculation with values of $t_f'$ from 30 to 200 days and found the optimum values at each theta level (see Section S2 of the supplement), which is why $f$ has a $z$ dependence in Equations 4 and 5. The values of $t_f'$ we use range from 50 days below 380 K

to 150 days above 400 K. The fraction of air from each region that contributed to the sampled UTLS can be calculated by

$$F_{NA}(K,z) = \int_0^\infty G_{NA}(K,z,t')dt' \tag{6}$$

where $F_{NA}$ refers to the fraction of air from the NA surface and $F_{NA} + F_{TR} = 1$.


Within each of the tropical and NA surface source regions we consider that the air may have come preferentially from a particular latitude band within each region. As seen in Figures 2 and S2 the surface trace gas time series can have significant latitudinal gradients so that the BL mixing ratios can vary widely within the tropics and NA. To allow for the calculation to identify optimal latitudinal source regions within the tropics and NA, we divide each region into subregions represented by

latitudinal Gaussian distributions ($L_{TR}, L_{NA}$) as shown in Figure S4. We then convolve the $L_{TR}$ and $L_{NA}$ distributions with the surface trace gas latitudinal distributions at each time $t'$ by

$$\chi_{iTR}(t',y_{TR}) = \int_{y_1}^{y_2} \chi_i(y_o, t-t')L_{TR}(y_o, y_{pTR})dy_o \tag{7}$$
$$\chi_{iNA}(t',y_{NA}) = \int_{y_1}^{y_2} \chi_i(y_o, t-t')L_{NA}(y_o, y_{pNA})dy_o \tag{8}$$


where $y_o$ is the surface latitude, $y_1$=20°S, $y_2$=70°N and $y_{pTR}$ and $y_{pNA}$ represent the peak surface latitudes of the distributions in the tropics and NA, respectively. These latitudinally convolved BL trace gas time series representing the subregions within the tropics and NA are then convolved in time with the age spectra by

$$\chi_{ioTR}(K,z,y_{TR}) = \int_0^\infty \chi_{iTR}(t', y_{TR})G_{TR}(K,z,t')dt' \tag{9}$$
$$\chi_{ioNA}(K,z,y_{NA}) = \int_0^\infty \chi_{iNA}(t', y_{NA})G_{NA}(K,z,t')dt' \tag{10}$$

Note that $\chi_{ioTR}$ and $\chi_{ioNA}$ are scaled BL concentrations since the age spectra from each region, $G_{TR}$ and $G_{NA}$, are scaled by the fraction of air from each region $F_{TR}$ and $F_{NA}$. The actual BL mixing ratios from each region can be found by $\chi_{ioTR}/F_{TR}$ and

$\chi_{ioNA}/F_{NA}$. The BL trace gas mixing ratios from both regions are then given by

$$\chi_{io}(K,z,y_{TR},y_{NA}) = \chi_{ioTR}(K,z,y_{TR}) + \chi_{ioNA}(K,z,y_{NA}). \tag{11}$$

The values of $\chi_{io}$ represent the range of possible source mixing ratios to the NAM UTLS from the tropical and NA surface with

transport times parameterized by $K$.

From the BL mixing ratios $\chi_{io}$ we can rearrange Equation 1 and express the idealized BL fractions as

$$\mu_i(K,z,y_{TR},y_{NA}) = \frac{\chi_i(x,t)}{\chi_{io}(K,z,y_{TR},y_{NA})} = \int_0^\infty e^{-t'/\tau_i(x,t')}G(K,z,t')dt'. \tag{12}$$


The measurement-based BL fractions, $\mu_i^*$, will have the same dependencies and can be expressed as

$$\mu_i^*(K,z,y_{TR},y_{NA}) = \frac{\chi_i^*(x,t)}{\chi_{io}(K,z,y_{TR},y_{NA})} \tag{13}$$

where $\chi_i^*(\boldsymbol{x},t)$ is the measured mixing ratio of trace gas $i$ at location $\boldsymbol{x}$ and time $t$.

To encompass the full range of possible age spectra in the UTLS we allow the calculation to optimize among a wide range of $K$, $y_{TR}$ and $y_{NA}$ indices at each theta level (see Section S4 of the supplement). The mean ages of the possible spectra vary from several months to several years and modal ages are as young as one day. The tails of the age spectra $G$ are extended to 30 years

so that is the upper limit used in the integrations.



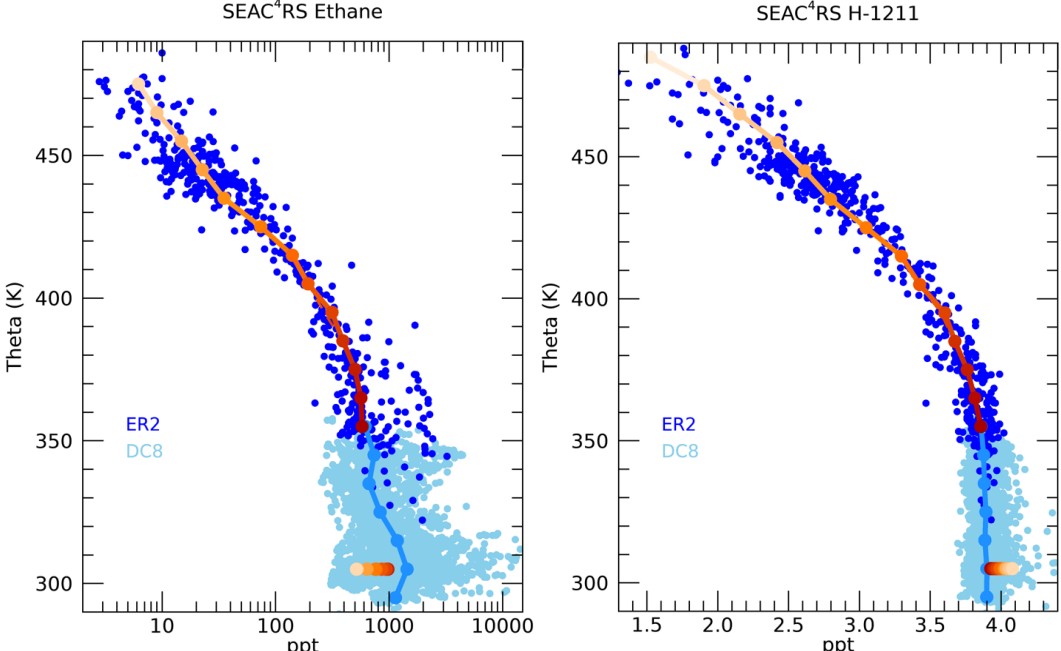

**Figure 4.** *Profiles of ethane and halon-1211 during the SEAC⁴RS mission as measured by the WAS instrument on the ER-2 (dark blue circles) and on the DC-8 (light blue circles). Theta averages for each 10 K interval are indicated by the solid lines with dark to light orange symbols representing the theta level for the ER-2 measurements and light blue for the DC-8. The boundary layer values ($\chi_{io}$) are indicated by the circles at 305 K with orange shading corresponding to the associated theta level in the ER-2 average profile.*

In the first part of this analysis, we use the average theta profiles of the trace gases between 350-480 K measured during SEAC⁴RS. These average theta profiles provide a robust set of measurements to establish the validity of the calculation. Profiles of ethane and halon-1211 are shown in Figure 4 as examples of the average profiles for these trace gases and the spread in individual measurements around the averages. We only use the ER-2 WAS measurements above 350 K and averages are taken over each 10 K layer. The BL mixing ratios $\chi_{io}$, based on the optimized $K$, $y_{TR}$ and $y_{NA}$ as described below, for each theta layer are shown by the colored symbols at 305 K. Note that for a trace gas in temporal decline, such as halon-1211, the BL mixing ratios from higher theta levels and older ages will have larger values since the air will have originated from the surface at a time when halon-1211 mixing ratios were larger than during the time of the mission. Ethane is also in decline but as shown in Figure 2, the NA BL mixing ratios are much larger than the tropical BL mixing ratios. In this case, the higher theta levels also correspond to a more tropical source of air with low enough mixing ratios compared to the NA source that this shift in source region offsets the decline in time. Thus, the ethane BL mixing ratios for higher theta levels are smaller compared to those from lower theta levels, just the opposite as for halon-1211.

We start with the calculation of $\mu$ and $\mu^*$ in the lowest theta layer (350-360K) using the UT lifetimes shown in **Table 1** as initial estimates of $\tau_i$. Following Hauck et al. (2020), we use the symmetric signed percentage bias (SSPB) (Morley et al., 2018) as a metric of the differences between $\mu^*$ and $\mu$ values for every $K$, $y_{TR}$, $y_{NH}$ combination. The SSPB can account for, and more evenly weight, differences over a large range of values better than a root mean square. We also utilize measured $CO_2$ mixing ratios ($\chi_{CO2}^*$) as an additional constraint to compare to $CO_2$ calculated from BL time series and the $K$, $y_{TR}$, $y_{NH}$ combinations using Equations 7-11 ($\chi_{CO2o}^*$). A simple mixing ratio difference is used to compare the measured and calculated $CO_2$. A combined difference quantity ($D$) is used to determine the optimum $K$, $y_{TR}$, $y_{NH}$ indices, and thus age spectra and surface source regions, at each theta level (see Section S4 of the supplement). When a minimum $D$ and corresponding age spectrum is found for a theta layer, the $\tau_i(z)$ values are adjusted to match the best fit idealized $\mu - \tau$ relationship. These $\tau_i(z)$ values are then used as the initial $\tau_i(z + 1)$ estimates in the optimization for the next vertical level.

As an example of the method, Figure 5a shows the $\mu - \tau$ relationships and age spectra for the 380 K level. A range of $\mu - \tau$ relationships and associated age spectra are shown along with the $\mu^* - \tau$ values from the measurements over a range of source region combinations. The initial lifetimes associated with each of the $\mu^*$ values are based on those found at the 370 K level. The





best fit based on the minimum value of $D$ is shown by the dark blue line and the adjusted $\tau_i$ are shown in red. Those trace gases primarily destroyed by OH in the tropical troposphere generally have increased $\tau_i$ values after the adjustment, and those trace gases primarily destroyed by photolysis generally have decreased $\tau_i$. This is consistent with pathways to a higher theta level in the UTLS encountering longer OH local lifetimes and shorter photolytic lifetimes. The six trace gases in the inset plot that

decrease in lifetime are all photolytically destroyed in the stratosphere. The corresponding best fit total age spectra is shown in the dark blue line and the age spectra from the tropical and NA surface are shown in Figure 5b. The age spectra from the different source regions have very different modal and mean ages. After an initial iteration of the method was performed, an adjustment was made to the average theta profiles following the calculation with the individual measurements as described below and in Section S1 of the supplement. This adjustment is especially important for the shortest-lived trace gases, such as

ethane shown in Figure 4, which have a number of highly elevated mixing ratios below 400 K due to pollution sources as described in Section 4.2.

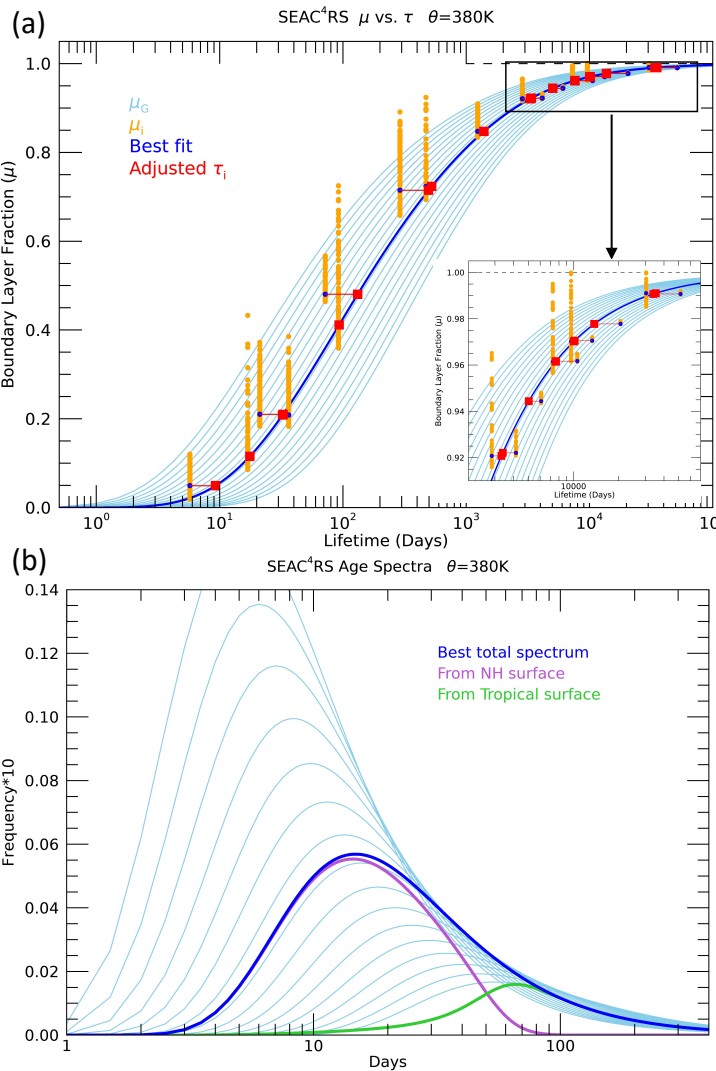


*Figure 5. (a) BL fractions calculated for a set of idealized trace gases (μ) from age spectra with different values of the parameter K (light blue lines in (a) and (b)) and from a set of 20 different trace gases measured during SEAC⁴RS and averaged in the 380-390K layer (μ*) (orange circles). The μ* values are only shown for the BL mixing ratios (χ_{io}) calculated with the K value and age spectra with the best fit at this level (blue line). The blue circles show μ* values with the source region combination that best fits the idealized μ − τ as well as the measured CO₂ (see text). The τ_i values are then adjusted so that each μ* value falls*




*exactly on the best fit $\mu - \tau$ curve (red squares). The inset plot in the lower right corner is an expansion of the lifetime range older than 2000 days. (b) Age spectra with the same range of K values as shown in (a), and the best fit spectra (blue) corresponding to that shown in (a). The age spectra from the NA surface (25°-60°N) (light purple) and the tropical surface (25°S-25°N) (light green) are also shown.*


## 4 Results

### 4.1 Average theta profiles

With the methods described above applied to the average theta profiles of SEAC⁴RS measurements, we derive a range of UTLS transport characteristics in the summer NAM region. The $\mu - \tau$ relationships, age spectra and source latitude distributions for all the theta layers are shown in Figure 6 and the path-integrated lifetime profiles of each trace gas are shown in Figure 7. This combination of transport quantities has not previously been derived from in situ UTLS trace gas measurements in any study that we are aware of. Figure 6a includes the $\mu^* - \tau$ relationships based on the idealized trace gases and the $\mu^* - \tau$ relationships in
the colored symbols. Each of the measured trace gases follows a unique path as a function of theta on the $\mu - \tau$ plot based on destruction source and growth history.

The age spectra from the NA and tropical surfaces (Figure 6b,d) show the change from rapid time scale (days), NA influence in the lowest theta layers, to long time scale (months to years) tropical influence in the higher theta layers. The NH age spectrum
for the 350-360 K layer has a modal age of ~2.5 days, similar to the 2 day modal age derived from measurements in the UT of the convectively active tropical Pacific [Luo et al., 2018]. The modal age from the tropical surface is ~2 months in the lowest theta layers and 3-4 months in the highest layers. Profiles of the modal and mean ages from the theta average results are shown in Figure 10 in the following section on the individual measurement calculation.

The surface source latitude distributions for each theta layer reveal the expected transition from local influence in the lower theta layers, below 400 K, to deep tropical surface influence in the higher theta layers above 420 K. The NA source distributions peak in the 40-50°N range for the sampled UT and move south to 30-40°N in the tropopause region and the LS. The distributions are scaled by the NA and tropical source fractions, $F_{NA}$ and $F_{TR}$, profiles of which are shown in the next section in Figure 10. NA source fractions are 0.4-0.5 below 370 K so the peaks at 40-50°N are of similar size to the tropical peaks, while above 420 K the
NA fractions are less than 0.2 so the tropical peaks at 0-10°N are dominant for those layers.

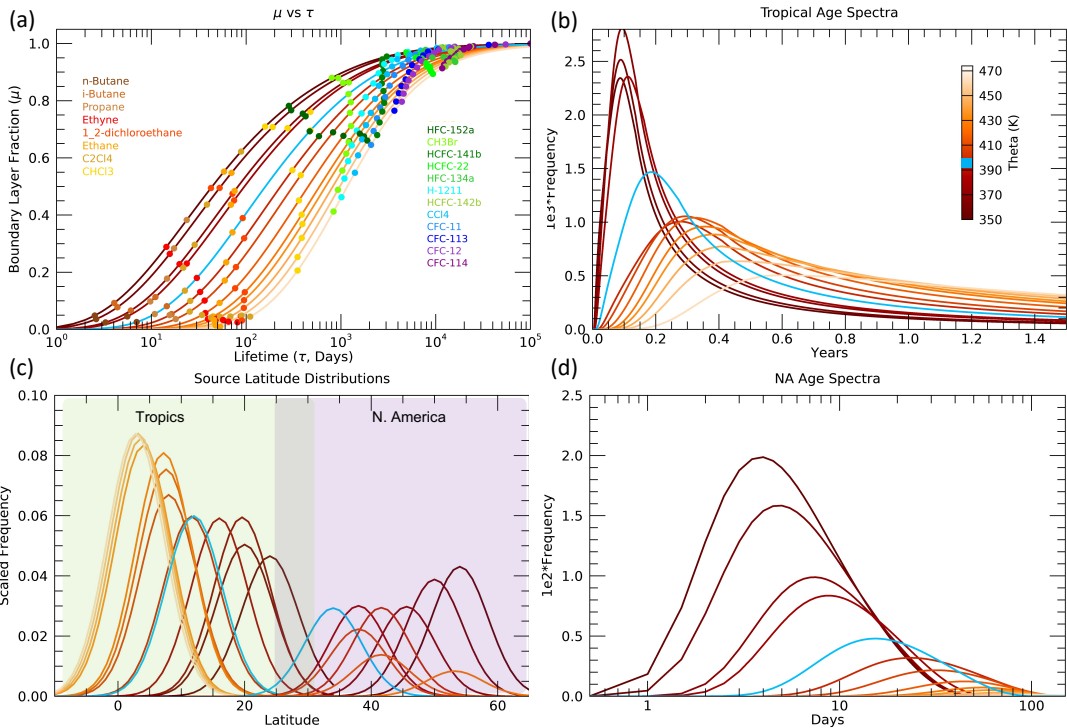



*Figure 6. (a) Boundary layer fractions (μ) vs. path-integrated lifetimes (τ) for each theta layer from 350-470 K in 10 K intervals. The solid lines are the μ − τ relationships based on the best fit age spectra shown in (b) and (d) and as described in the methods section. The colored symbols are the μ\* − τ relationships based on the 20 trace gases listed in the legend and Table 1. (b) Tropical and (d) NA source region best fit age spectra for each theta layer. (c) Tropical and NA source region distributions associated with the age spectra and used to calculate the μ\* values shown in (a). The distributions are scaled by the total tropical vs. NA source region contribution in each theta layer. The light blue lines in each plot represent the distributions for the 390-400 K layer.*

The path integrated trace gas lifetimes consistent with the age spectra and source latitude distributions in Figure 6 are shown in Figure 7. As described in the Methods section, the initial values of $\tau_i$ in the lowest 350-360 K layer are listed in Table 1 for each trace gas. However, when an optimum value of $K$ is found, the $\tau_i$ values are adjusted so that the values shown in Figure 7 do not necessarily match those in Table 1 in the 350-360 K layer. In each subsequent higher theta layer, the $\tau_i$ values are adjusted based on the optimal $\mu − \tau$ relationship in that layer. The $\tau_i$ profiles reveal a number of interesting features that are generally consistent with what would be expected based on the different sink mechanisms and locations for each of the trace gases.

The shortest-lived trace gases have $\tau_i$ values that mostly increase with theta, as expected since they react with OH radicals that decrease with altitude caused by the decrease of water vapor. As the sampled air parcels move further from the troposphere in higher theta layers, the pathways of transport from the surface to those theta layers will include more regions with less OH and longer local lifetimes. The exceptions are propane and perchloroethylene ($C_2Cl_4$) below 400 K. The $\tau_i$ profiles for the short-lived trace gases can be influenced by the amount of tropical vs. extratropical sources since their local lifetimes vary significantly with latitude. Propane, ethyne, ethane and the butanes have local lifetimes estimated to be roughly a factor of two longer in the NH subtropics compared to the tropics (Luo et al., 2018; Tang et al., 2007). But the details of how the local lifetimes of these trace gases vary with latitude and season are not well known so it is difficult to clearly attribute the propane and perchloroethylene $\tau_i$ decreases below 400 K.

The HFCs and HCFCs also have increasing values of $\tau_i$ with higher theta due to the dominance of OH destruction in the troposphere for these species (WMO, 2018). Therefore, the $\tau_i$ profiles of these species generally have relatively smaller values at the lowest theta levels compared to the CFCs, and cross over to have relatively larger values than the CFCs at the highest theta levels where the photolytic destruction of CFCs becomes the dominant loss process. The trace gas with the longest stratospheric lifetime of any used in this study is HFC-134a at 267 years [WMO, 2018] and it has the largest value of $\tau_i$ in the 470-480 K layer at ~$3.5\times10^4$ days or ~100 years. In the 350-360 K layer the $\tau_i$ value for HFC-134a is ~$3.5\times10^3$ days or ~10 years, so the $\tau_i$ value for this trace gas increases by an order of magnitude over the sampled UTLS region. The 10-year value of $\tau_i$ for HFC-134a in the UT is consistent with the local lifetime estimates of Chelpon et al. (2021) of ~5 years in the tropical troposphere and 15-20 years over the whole troposphere (WMO, 2018; Chelpon et al., 2021).

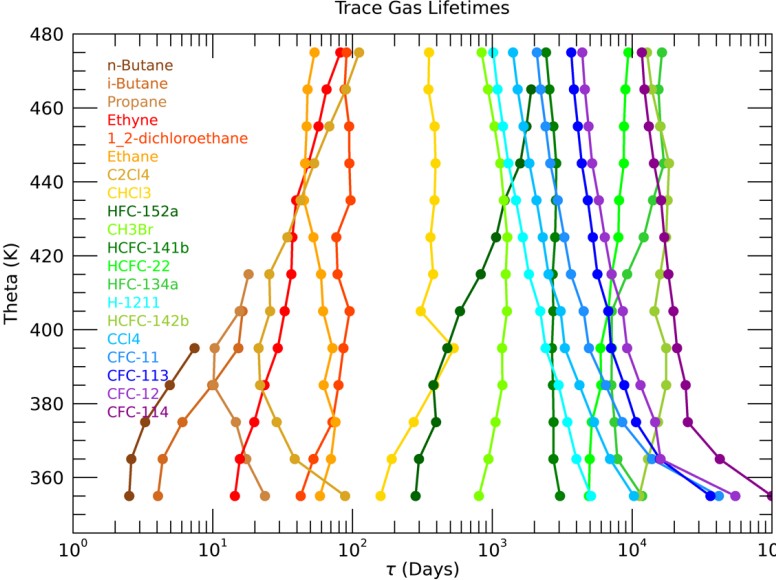

*Figure 7. Profiles of the path-integrated trace gas lifetimes ($\tau_i$) derived from the theta average profiles.*



The CFCs are destroyed by photolysis in the stratosphere so all of their $\tau_i$ values are very large in the UT and decrease with increasing theta as more transport pathways pass through the stratospheric loss regions. Halon-1211 has loss from both OH in the troposphere and photolysis in the UTLS so it has a $\tau_i$ profile that decreases less rapidly than the CFCs (Chipperfield et al., 2013; WMO, 2018). Chloroform (CHCl$_3$) and methyl bromide (CHBr$_3$) also have both OH and photolytic loss but much shorter tropospheric lifetimes (2-3 months and 2 years) relative to their stratospheric lifetimes (1-2 years and 33 years) compared to

halon-1211 (10-15 years and 30-40 years) (Saltzman et al., 2004; Chipperfield et al., 2013; WMO, 2018). This ratio of stratospheric to tropospheric lifetimes of ~10 results in $\tau$ profiles of chloroform and methyl bromide with increasing values below 400 K as for the short-lived trace gases, but decreasing values above 400 K similar to the CFCs.

The contrast between the $\tau_i$ profiles of chloroform and HFC-152a clearly illustrates the effect of trace gases with somewhat

similar tropospheric lifetimes but much different stratospheric lifetimes. The $\tau_i$ values for these two trace gases are basically identical in the 380-400 K layer but HFC-152a $\tau_i$ values increase above this level while those of chloroform decrease. The difference in $\tau_i$ reaches a factor of three above 460 K and this reflects the relative stratospheric lifetimes of 33 years for HFC-152a compared to 1-2 years for chloroform [WMO, 2018].

It is important to note that features of the $\tau_i$ profiles described above were not prescribed ahead of time in the optimization method. An initial constraint on the $\tau_i$ values was required for the lowest theta layer but even in that layer the $\tau_i$ values were allowed to vary based on the best fit $\mu - \tau$ relationship. Thus, while the absolute values of $\tau_i$ have some uncertainty, the relative values and theta profile shapes are robust and help confirm the validity of the method as well as all of the surface time series in relation to the ER-2 UTLS measurements.


### *4.2 Individual Measurement Locations*

Following the calculation of age spectra and other transport diagnostics based on the mean theta profiles, we performed the same method on each set of measurements taken at the individual sampled locations as shown in Figures 1 and 3. A minimum of 10

species from Table 1 must have been measured from WAS at a location for the calculation to be performed. We initialize these calculations with the age spectra and trace gas lifetimes derived from the average theta profiles for the layers in which the individual measurements were taken. As will be shown below, the calculation with the average theta profiles is essential to provide context and a means to accommodate into the method the wide range of individual measured mixing ratios.

An immediate issue with the individual measurements is the occurrence of UTLS mixing ratios larger than the BL mixing ratios, which would result in $\mu^* > 1$. This violates the assumption of the method as described by Equation 1 where $\mu = 1$ is only defined for $t' = 0$ in the boundary layer. Equation 1 excludes the case where $\mu > 1$. The range of measured UTLS mixing ratios is expected based on the range of measured mixing ratios in the free troposphere and BL by the DC-8 as shown for example for ethane in Figure 4 and the butane species in Figure S1. Our assumption is that the relatively large mixing ratios below 420 K,

especially when $\mu^* > 1$ but even when $\mu^* < 1$ but falls outside the expected $\mu - \tau$ relationship, are primarily driven by NA source variability since transport alone cannot cause such high values. Above 420 K, the spread is primarily driven by transport variability since the NA BL has a relatively small influence at those locations.

A strength of this method, including the wide range of trace gases used here, is that outliers from the $\mu - \tau$ relationships are

readily apparent and the magnitude of adjustments necessary to bring the outliers in line with the other trace gases can be well approximated. We make the assumption that outliers are the result of 'polluted' source regions and so we perform several steps to identify and adjust the boundary conditions with a scaling factor $S$ for the trace gases with enhanced UTLS mixing ratios. The result of this scaling is that essentially all of the available measurements can be used in the calculation and we are able to quantify a range of 'polluted' source regions to the NAM UTLS.


To begin to address the outlier measurements with relatively large mixing ratios we normalize the individual mixing ratios with the mean profiles of each trace gas by $\chi_{inorm}(x) = \chi_i(x)/\chi_i(z)$ where $x$ refers to an individual measurement location (Figure S7). All of the $\chi_{inorm}$ profiles have a similar pattern of variability with the largest spread of values below 420 K and a much smaller spread above that level. Our assumption is that the spread below 420 K is primarily driven by NA source region

variability since transport alone cannot cause such high values, and above this level the spread is primarily driven by transport variability since the NA has a relatively small influence at those locations. Thus, we define a scaling factor for $\theta(x) < 420K$,

$$S_{inorm}(x) = \begin{cases} 1 & , \ \chi_{inorm}(x) < 1.2 \\ \chi_{inorm}(x), & \chi_{inorm}(x) \geq 1.2 \end{cases} . \tag{14}$$

That is, for a trace gas with a mixing ratio 20% or more larger than the theta layer average value, we scale $\chi_{io}$ by $S_{inorm}$ so that ideally all $\mu^*(K, z, y_{TR}, y_{NH}) < 1$. We find initial estimates of $\mu_i^*(x)$ using the optimized $\chi_{io}$ BL values from the theta average layer that encompasses the individual measurement location.



After the initial scaling we address remaining $\mu^*(\boldsymbol{x}) > 1$ outliers with an additional scaling factor

$$S_{i\mu}(\boldsymbol{x}) = \begin{cases} 1 & , \quad \mu_i^*(\boldsymbol{x}) \leq 1 \\ \mu_i^*(\boldsymbol{x})/\mu_i^*(z), & \quad \mu_i^*(\boldsymbol{x}) > 1 \end{cases} \tag{15}$$

where $\mu^*(z)$ is the BL fraction for the theta average layer (Figure 6a) that encompasses the measurement location $\boldsymbol{x}$. A combined scaling factor defined as $S_i = S_{inorm} * S_{i\mu}$ is then applied to all $\chi_{io}$ and an initial optimization is performed as was
done for the theta average profiles in the previous section. For each location the trace gas lifetimes are initialized with the theta layer average values $\tau_i(z)$ and adjusted following the optimization to the $\tau_i(\boldsymbol{x})$ values.

After this initial optimization we compare the $\tau_i(\boldsymbol{x})$ to $\tau_i(z)$ values and make adjustments to $\chi_{io}$ for any trace gas with $\tau_i(\boldsymbol{x})$ that is a factor of 4 smaller or larger than $\tau_i(z)$. That is,

$$S_{i\tau}(\boldsymbol{x}) = \begin{cases} 1 & , \quad \tau_i(z)/4 \leq \tau_i(\boldsymbol{x}) \leq 4 * \tau_i(z) \\ \mu_i^*(\boldsymbol{x})/\mu_i^*(z), & \quad \tau_i(\boldsymbol{x}) < \tau_i(z)/4 \text{ or } \tau_i(\boldsymbol{x}) > 4 * \tau_i(z) \end{cases}. \tag{16}$$

The reason we make this adjustment is that even with the scaling described above there are still outliers in $\mu_i^*(\boldsymbol{x})$ that are apparent in the $\mu - \tau$ relationships. We do not want to overly constrain the possible transport variability revealed by the
majority of individual measurements within a theta layer so the initial scaling steps are only intended to identify clear outliers and broadly bring the trace gases into $\mu - \tau$ alignment. Outliers that remain following the initial optimization will generally not affect the best fit and value of $K$, but will appear as outliers in the profiles of $\tau_i(\boldsymbol{x})$. The overall scaling factor is then defined as $S_i = S_{inorm} * S_{i\mu} * S_{i\tau}$. After the scaling factor $S_{i\tau}$ is applied to $\chi_{io}$ the optimization is performed again to calculate the final transport parameters for each individual location.

An example of the BL scaling and age spectra optimization for an individual location with a relatively 'polluted' BL source for certain trace gases is shown in the $\mu - \tau$ relationships in Figure 8. There is a wide spread in the initial $\mu_i^*(\boldsymbol{x})$, with values greater than 1 for a range of trace gases from propane and ethane to HFCs and CCl$_4$, while several other trace gases, such as 1,2-dichloroethane, CHCl$_3$ and CFC-113, have values close to the $\mu - \tau$ relationship for the theta layer average indicated by the
dotted line.

Following the $S_{inorm}(\boldsymbol{x})$ scaling, the $\mu^*(\boldsymbol{x})$ values decrease for many of the short-lived trace gases, as indicated by the upward triangles (Figure 8a), and there are no remaining values of $\mu^*(\boldsymbol{x}) > 1$ for these species. However, for the longest-lived trace gases (Figure 8b) the $S_{inorm}(\boldsymbol{x})$ scaling has no effect because the condition of $\chi_{inorm}(\boldsymbol{x}) \geq 1.2$ was not met for any of these
species (the upward triangles are not shown since there is no change from the initial values). For the longest-lived species the $S_{i\mu}(\boldsymbol{x})$ scaling (left facing triangles) brings the elevated $\mu^*(\boldsymbol{x})$ values below 1 so the optimization can be performed. The left facing triangles are not shown for the shortest-lived species since there is no change from the $S_{inorm}(\boldsymbol{x})$ scaling. Three of the trace gases (propane, HCFC-22 and CFC-11) are affected by the $S_{i\tau}(\boldsymbol{x})$ scaling (upside down triangles) since they fall in the green shaded area of Figure 8 which indicates $\tau_i$ is outside the range expected for this theta layer.





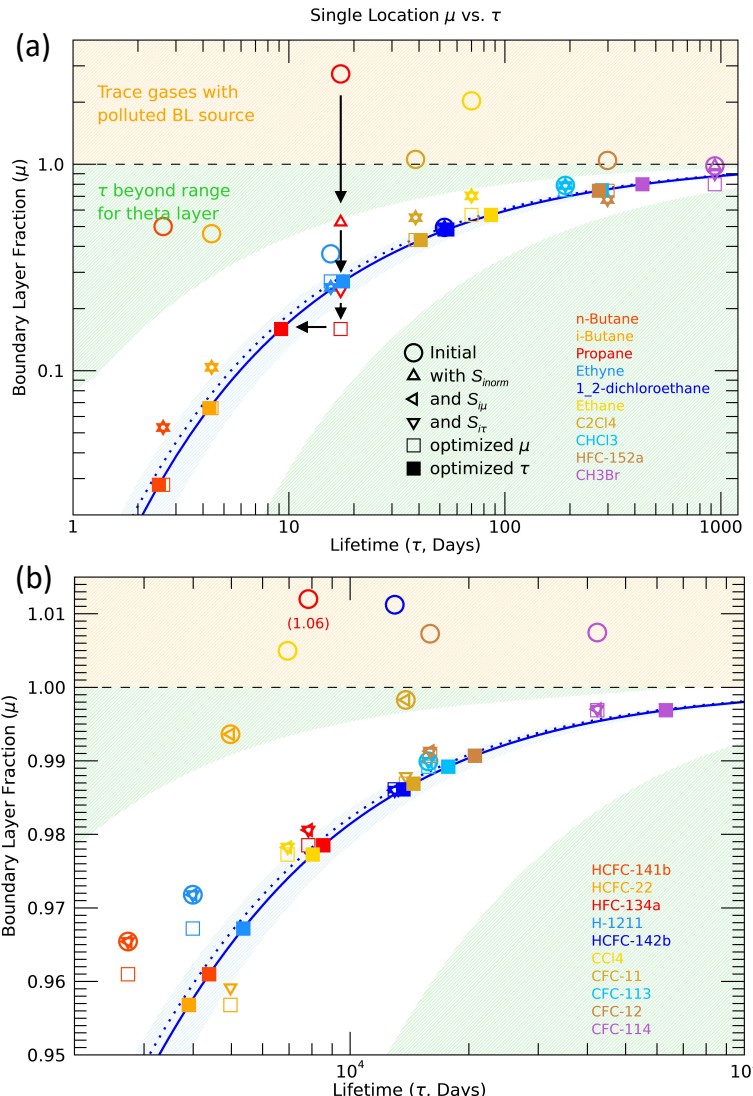

*Figure 8.  Boundary layer fraction (μ) vs path-integrated lifetime (τ) for a single SEAC⁴RS measurement location near the tropopause (θ=362 K, 29°N).  The 10 shortest-lived species are shown in (a) and the 10 longest-lived species with a narrowed μ range are shown in (b).  The orange shaded region where μ > 1 indicates a polluted BL source region, while the green shaded region indicates where the estimated τ values fall outside the expected range for the theta layer.  The μ − τ relationship for the theta layer average is indicated by the dotted blue line and the best fit for this individual location by the solid blue line.  The symbols represent adjustments made to μ*(x) by the various scaling factors applied to the boundary conditions χᵢₒ and from the optimization (see text).*

For the location shown in Figure 8 it could easily be assumed that with relatively large measured mixing ratios of short-lived trace gases such as n-butane, propane and ethane the dominant age of the sampled air parcel would be very short, essentially hours or days since it was convectively transported to this UT location. However, with this method we find an age spectrum with a modal age of 5 days and a mean age of roughly half a year for this location. These time scales are necessary to explain the $\mu^* - \tau_i$ relationships of trace gases such as 1,2-dichloroethane, CHCl₃, halon 1211 and CFC-113, which lie almost exactly on the $\mu - \tau$ curve for the theta layer average. There is no way that trace gases such as these with $\tau_i$ ranging from several months to 50 years could have values of $\mu^*$ significantly less than 1 as they do, without frequency in the age spectra representing time



scales from weeks to years. There is the possibility that the $\tau_i$ values for these four trace gases are overestimated or their BL
time series are overestimated such that their $\mu^*$ values are too low. But different trace gases exhibit the average theta layer $\mu - \tau$
relationship for different air parcels and the initial calculation with the average theta profiles establishes the framework among
the trace gases that would reveal systematic errors in BL time series, for example. So, while this air parcel does have a
component of the age spectra representing transport time scales of hours to days, this is relatively small compared to the longer
weeks to months transport time scales. And the explanation for the relatively high measured mixing ratios of a number of short-
lived trace gases that becomes clear with this method, is that the air parcel was influenced by transport from polluted BL
locations with highly elevated levels of certain trace gases.

*Figure 9. Boundary layer fractions (μ\*) vs. theta for propane and ethane for the theta profile averages (solid lines), initial*
*estimates of μ\* (open circles) and final estimates of μ\* (solid squares) following scaling of BL time series and optimization*
*calculation (see text). Path-integrated lifetime (τ) profiles for propane and ethane based on the average theta profiles (solid*
*lines, open circles) and the individual measurement locations (solid squares). The constraint limits used in the calculation of the*






*individual location age spectra are shown by the dotted lines representing lifetimes a factor of four smaller or larger than the average profile.*

Examples of the individual measurement location outlier identification technique are shown in the profiles of $\mu^*$ and $\tau$ for propane and ethane (Figure 9). Both propane and ethane initially have a significant number of values of $\mu^* > 1$ (open circles), as expected from the normalized mixing ratios shown in Figure S7. Following the scaling of outliers and optimization for the individual measurements, the final estimates have no values of $\mu^* > 1$ and much less spread around the theta average profile,

especially on the high end of the values. The $\tau$ profiles have essentially the opposite behavior between the initial and final estimates since we initialize the individual locations with the values from the theta average profile. This means all of the individual trace gases in a theta layer will initially have the theta average $\tau_i(z)$ values. After the scaling and optimization, the individual $\tau_i(x)$ values are estimated from the best fit $\mu - \tau$ curves. This introduces a spread in the $\tau_i(x)$ values that is constrained to be within a factor of four greater or smaller than the $\tau_i(z)$ values. For the longest-lived trace gases this spread is

much less than a factor of four. Almost all of the values of $\tau_i(x)$ lie within the limits shown by the dotted lines in Figure 9 with the exception of a couple of propane values. In these cases, the propane BL time series could not be scaled effectively and they remain outliers. But in nearly all locations the $\tau_i(x)$ are able to be constrained for all of the trace gases. The constraint limits are arbitrary but after some experimentation these limits appear to account for clear outliers while also allowing variability within each theta layer that is expected due to transport variability.


In the calculation of $\mu^*$ as shown in Equation 13, the scaling was performed on the total $\chi_{io}$ values. But what this scaling implies is that there is a relatively 'polluted' surface region that impacted the measured mixing ratios in the UTLS. The polluted surface region was almost certainly in NA for two main reasons. One is that we know there are relatively large mixing ratios of all of the trace gases in this study in the lower troposphere over NA as measured during SEAC⁴RS (Figure 4 and S1) that come from

various pollution sources. And two, while there are significant pollution sources in the tropics, they will most likely be at longitudes far from where SEAC⁴RS flights sampled the UTLS, such as Asia, and thus would be mixed with the background tropical troposphere before entering the sampled UTLS.

Assuming the pollution sources that affected the sampled UTLS during SEAC⁴RS were from the NA only, the scaling factor

should only be applied to the $\chi_{ioNA}$ in Equation 11. Since the scaling factors derived above relate to $\chi_{io}$, to apply a scaling only to $\chi_{ioNA}$, we need to increase the scaling factor by the inverse of the fraction of air from NA. That is,

$$S_{iNA}(x) = \frac{S_i(x)}{F_{NA}(x)} \tag{17}$$

where $S_i(x)$ is the scaling factor derived to apply to $\chi_{io}$ for trace gas $i$ and measurement location $x$, and $S_{iNA}(x)$ is the scale factor applied only to $\chi_{ioNA}$. As seen in Figure 10, $F_{NA}$ ranges from 0.2-0.6 for theta less than 410 K where the scaling is applied. Thus, the derived values of $S_i$ are increased by factors of 1.6-5 to obtain the values of $S_{iNA}$.

Following the scaling of $\chi_{ioNA}$, we compute optimized $K$, associated age spectra and surface source latitudes for each individual

WAS ER-2 measurement location. Profiles of the mean ages, modal ages and NA source fractions are shown in Figure 10. The mean ages range from several months in the UT to several years in the LS with much more spread in values at the higher theta levels primarily driven by a latitudinal gradient as seen in Figure 11. The modal ages range from days in the UT to months in the LS with a large latitudinal gradient above 450 K.

The values of $F_{NA}$ (Figures 10 and 11) range from 0.2-0.6 in the UT, that is 20-60% of the air in the sampled parcels originated from the NA surface north of 25°N, to less than 0.1 in the LS above 450 K. From 370-410 K the theta average $F_{NA}$ values are nearly constant at ~0.3. This is related to the transition of the source region scaling factor time scale, $t_f$, from 50 to 150 days over this theta range. As the value of $t_f$ increases, $F_{NA}$ will increase if the age spectrum remains the same since more of the air at ages less than $t_f$ will have come from the NA surface. However, since the mean age increases with theta the amount of the age spectra

in the 50-150 day range decreases with theta, which roughly offsets the increase in the value of $f(t)$ in that age range. The individual measurements show a wide range of values of $F_{NA}$ in the 380-400 K layer from 0.1-0.4. The relatively low values of $F_{NA}$ in this layer are consistent with horizontal intrusions of tropical air into the LS above the subtropical jet, typically due to Rossby wave activity in the Spring (Pan et al., 2009).



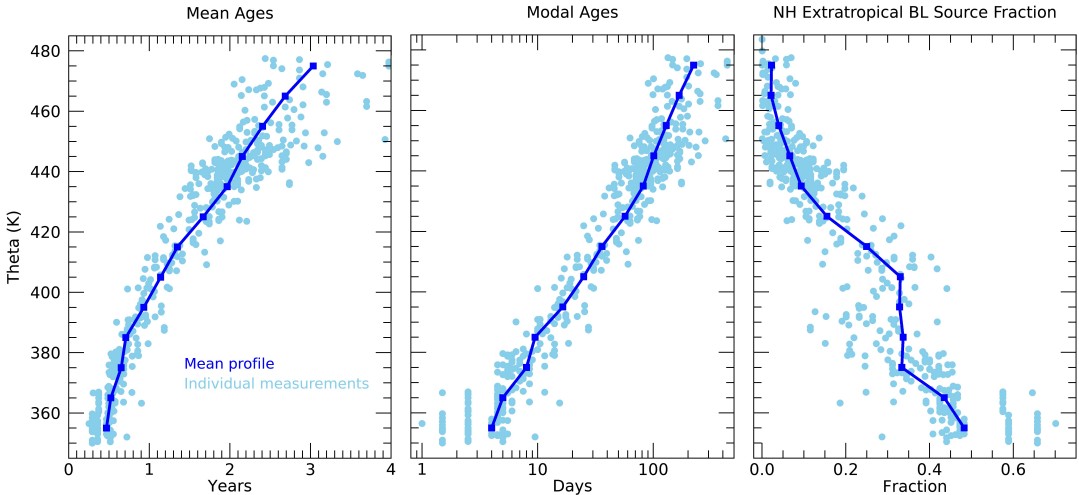


*Figure 10. Profiles of mean age, modal age and $F_{NA}$ as a function of theta. Results from the theta average profiles are shown in the dark blue lines and symbols and those from the individual measurement locations in the light blue symbols.*

Mean ages in the UTLS have been estimated from *in situ* measurements and model output in many previous studies (e.g. Boenisch et al., 2009; Diallo et al., 2012; Konopka et al., 2015; Ploeger and Birner, 2016; Hauck et al., 2020). The range of values calculated here are very similar to those from the same region and season in previous studies. For instance, in Ploeger and Birner (2016), CLaMS model output has mean ages during summer in the 20-40°N region that range from ~1-3 years between 350-500K. This provides some confidence that the mean age results shown here fall within expected values.

The modal ages can also be compared to previous estimates but this quantity is much less commonly shown over a range of latitudes and vertical levels and has not been calculated from the surface based on measurements that we are aware of. In Ploeger et al. (2019) the modal age based on CLaMS output from 20-40°N and 400 K in summer is 4-5 months, which is considerably longer than the 10-30 day modal ages from our calculation. The model results are based on zonal averages however so the convectively active NAM region would be expected to have shorter modal times due to the rapid convective 630 transport from the surface. A more recent study using CLaMS output (Yan et al., 2021) focused on transport from different surface latitude regions and found somewhat faster (~3 month) modal times for transport from the NH extratropical surface to the NH extratropical UTLS in summer. This study again only shows zonal mean results so it is likely that the modal times would be even shorter in the NAM region.

The $F_{NA}$ distributions can be compared to those from previous studies that calculated UTLS source region fractions from model output. In Orbe et al. (2015), the tropics were defined as 10°S-10°N and the NH extratropics north of 10°N, so the $F_{NA}$ values should be relatively large and the $F_{TR}$ relatively small compared to our results. That is roughly consistent with values of $F_{TR}$ in that study which range from 0.2 in the summer UT to 0.6 in the LS above 100 hPa, or ~400 K. And values of $F_{NA}$ from 0.7 in the UT to 0.3 in the LS. The main difference is the higher $F_{NA}$ values through the LS that likely relates to the wider source 640 region definition. Another study with this type of model surface source analysis is Yan et al. (2021) where the tropics were defined as 30°S-30°N so the $F_{NA}$ values should be relatively small compared to our results. In that study, $F_{NA}$ ranges from 0.06 in the NH summer UT to 0.02 in the LS, while $F_{TR}$ ranges from 0.94-0.97. These are clearly much smaller values of $F_{NA}$ compared to our results and it is not clear if the source region definitions are different enough to explain the discrepancy. Further comparisons of source region fractions defined from UTLS measurements and models would be a useful new diagnostic 645 of model transport and we intend to perform such analysis in a future study.



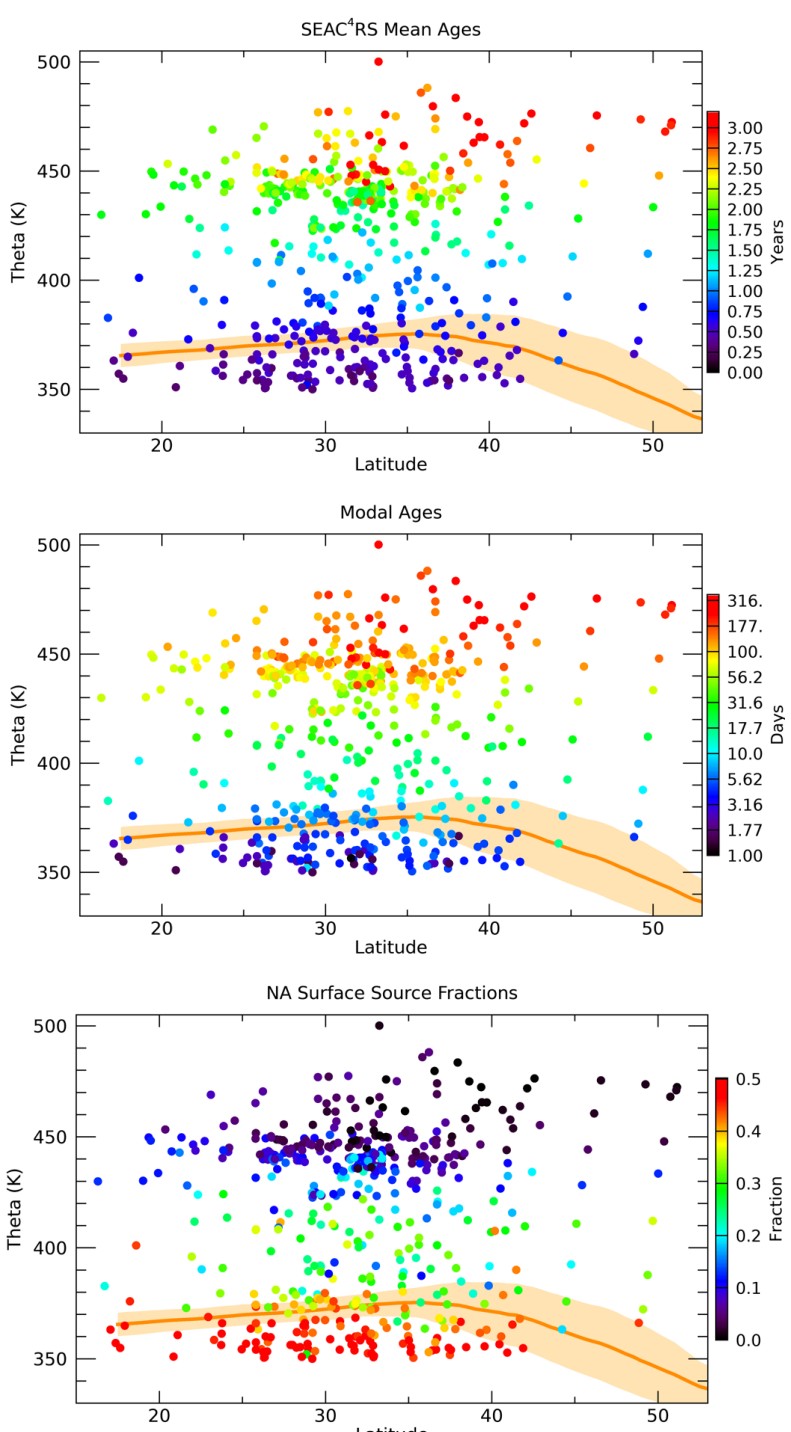

Figure 11. Latitude vs. theta distributions of the mean age, modal age and NA surface source fractions for each aircraft measurement location. Tropopause regions based on MERRA2 products are indicated by the orange shading and solid line.

The BL source peak latitudes ($y_{TR}$, $y_{NA}$) for the individual UTLS measurement locations are a unique aspect of this calculation (Figure S4). Figure 12 shows profiles of $y_{TR}$ and $y_{NA}$ scaled by the fractions of air from each region, $F_{TR}$ and $F_{NA}$. Only



locations with $CO_2$ measurements are shown since without those measurements the source latitudes are much less well constrained. The source region fractions show the largest contribution of the North American BL to the sampled UTLS is from 40°-55°N. The correspondence of the source latitudes to the $CO_2$ mixing ratios is shown in Figure 13. Below 370 K, the lowest $CO_2$ mixing ratios come from the highest latitudes, 50°-54°N. This follows from the large latitudinal gradient in $CO_2$ in the BL over NA during the summer as shown in Figure S2. Since the age spectra below 370 K are heavily weighted toward the most recent several weeks before the flights, only the highest latitudes have mixing ratios low enough during that time to account for 660 the lowest measured $CO_2$ mixing ratios. The higher the measured $CO_2$ mixing ratio, the further south the source fractions.

The tropical contributions are relatively small for theta lower than 370 K and spread across a range of tropical latitudes. The smallest symbols shown in Figure 12 have the most uncertainty since they represent small values of $F$ and thus have a small influence on the measured set of trace gases. The 370-400 K layer is a transition zone going from equal contributions from NA 665 and the tropics to a more tropical source. The wide range of source latitudes, especially in the 370-380 K layer, is reflected by the range of measured $CO_2$ mixing ratios in this part of the UTLS as shown in Figure 13. The NA source latitudes continue to be inversely related to the measured $CO_2$ as in the lower theta levels, while the tropical source latitudes continue to be less well correlated with the measured $CO_2$. This is mostly due to the much larger summertime latitudinal gradient of $CO_2$ north of 30°N compared to the tropical latitudes, which results in more leverage over the measured $CO_2$ by the high latitude source region 670 compared to the tropics. The largest measured $CO_2$ mixing ratio in the UTLS of 396.5 ppm at 380 K is shown to best fit a NA source peak latitude of 38°N and a tropical peak latitude of 20°N.

Above 400 K, there is a transition to tropical latitudes as the primary source region to the measured UTLS. The NA symbols are very small above 400 K and disappear altogether above 440 K which indicates values of $F_{NA} < 0.2$ at those locations. The 675 tropical source latitudes shift to the south, near the equator, with higher theta. Above 420 K the tropical source latitudes are between 4°S-16°N with the theta average profile at 8°N, roughly the position of the Intertropical Convergence Zone (ITCZ).

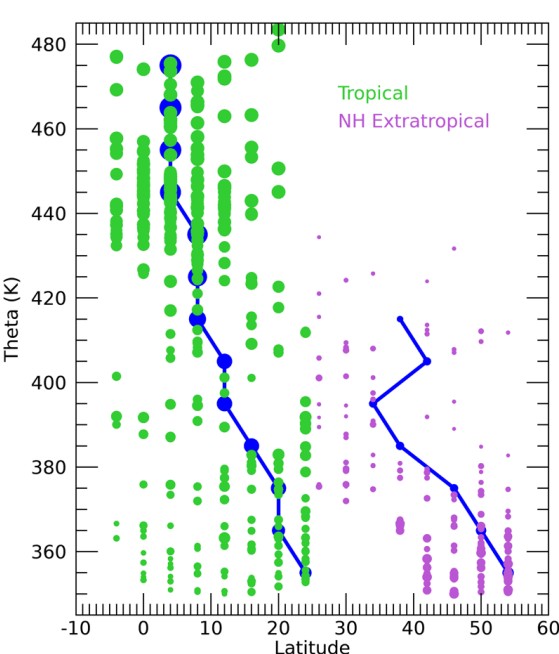

BL Source Peak Latitudes

*Figure 12. BL source peak tropical ($y_{TR}$, green) and NA ($y_{NA}$, purple) latitude profiles. The symbols are sized based on the*
*value of $F_{TR}$ for the tropical latitudes and $F_{NA}$ for the NA latitudes. The results for the theta average profiles are shown by the dark blue lines.*

We can also look at the distributions of the source latitudes from each region as a function of the measurement location in latitude and theta as shown in Figure 14. The symbols are sized as in Figures 12 and 13 to make the contribution from each 685 region at the different locations clear. The shift from northern subtropical to deep tropical source latitudes with increasing theta is apparent, as in the previous figures. There is no significant latitudinal gradient in the tropical source latitude within each theta





layer. Below 380 K, the largest sized symbols generally have source peak latitudes north of 20°N. This can also be seen in Figure 12 and generally matches the average theta profile tropical peak latitude of 20°N in the lower layers.

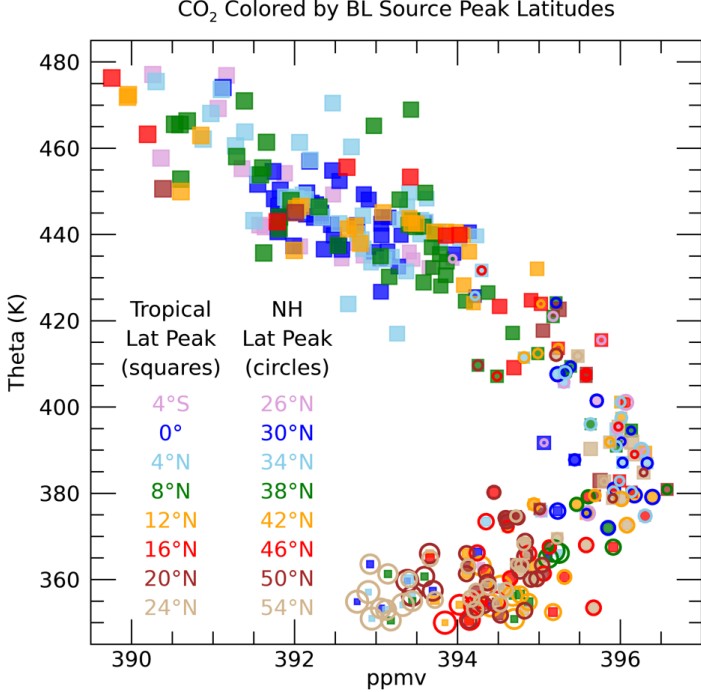


*Figure 13. Individual measurement CO₂ profile colored by the BL source peak latitudes in the tropics (squares) and NA (open circles). The symbols are colored by the source latitudes in each region as indicated by the legend, and sized by the values of $F_{TR}$ and $F_{NA}$ for the tropical and NA symbols respectively.*

The NA source peak latitude distribution shows the previously noted shift from high latitude sources in the lowest theta layers to lower latitude sources above 370 K where $F_{NA}$ becomes small. There is an interesting latitudinal gradient in the extratropical source latitudes below 370 K such that at the more southern sampled latitudes the NA sources are from the most northern locations. The anticyclonic circulation in the NAM UT region can transport air across a wide range of latitudes within days as seen in previous trajectory studies such as Herman et al. (2017), which was focused on the SEAC⁴RS mission. In that study,
convective overshooting regions were shown to influence sampled UTLS air masses up to a week or so later and 10-20° in latitude away.





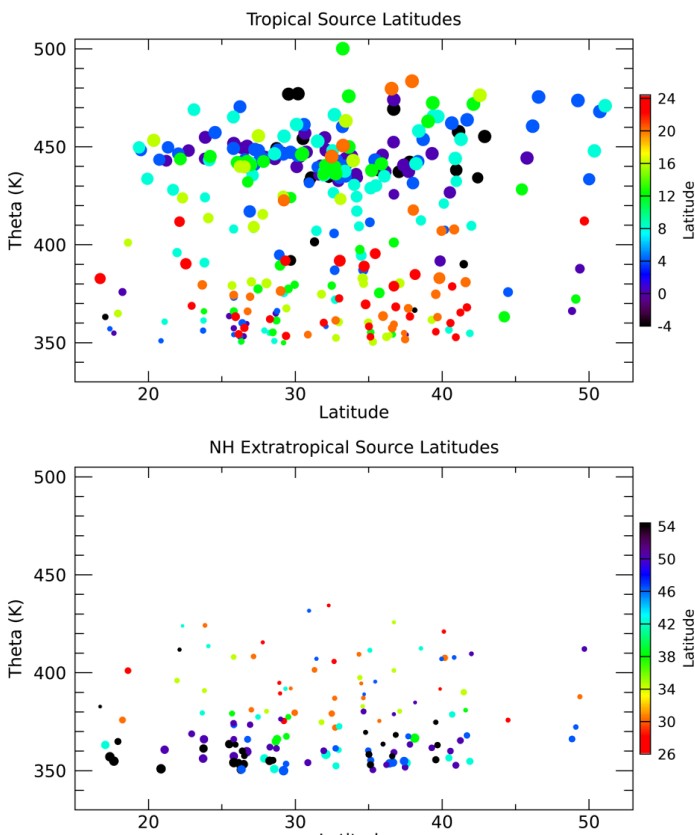

**Figure 14.** *Individual measurement locations as a function of latitude and theta colored by peak BL source latitude and sized by the values of $F_{TR}$ and $F_{NA}$ for the tropical and NA plots, respectively. Only locations where $F > 0.2$ and $CO_2$ measurements were available are shown in each plot.*

**Discussion**

The methods and results described here take advantage of the wide range of trace gas measurements taken during the SEAC[4]RS mission, as well as from surface sites around the world, to reveal a unique set of transport characteristics of the summertime UTLS over North America. This work builds on the techniques and ideas of many previous studies but especially on recent studies that have focused on maximizing the transport information derived from a suite of trace gas measurements (Luo et al., 2018; Hauck et al., 2019, 2020; Podglajen et al., 2019; Chelpon et al., 2021). This advancement in our knowledge of more detailed transport aspects of the UTLS from measurements is important to better constrain global climate models and reanalysis products in this region.

This work is ideally a step along the way towards more comprehensive measurement-based transport diagnostics utilizing the many other in situ and remote measurement data sets that exist today and those to come from future missions. The results shown here are by no means the exact answers for each transport diagnostic at each measurement location. We have made a number of assumptions in the method, which are described in the main text and the supplement, that could change individual results somewhat if different choices were made. But the overall patterns and self-consistency of the results are robust to a range of different assumptions from those made here.

An important aspect that has not been discussed thus far is the seasonal cycle in transport. We have incorporated the seasonal cycle in the trace gas mixing ratio time series in the boundary layer but we have not explicitly added a seasonal cycle to any of the transport diagnostics beyond what is revealed by the trace gas measurements. It has been shown in many previous modeling studies and in the recent work of Hauck et al. (2019, 2020) and Podglajen et al. (2019) that the seasonal cycle is a significant feature in the age spectra in the UTLS due to the seasonal cycle in transport in this region. These studies show that available trace gas measurements in the UTLS are generally not sensitive enough to the seasonal cycle to reveal the seasonal features seen



in age spectra from model output. A technique to account for the seasonal cycle revealed by models is to essentially parameterize a seasonal cycle into the age spectra and otherwise let the measurements define the rest of the spectra shape. We

did try this technique with our method and found that the results were very similar for all of the transport diagnostics. Since the focus of this study is on measurement-based diagnostics we decided to leave the model-derived seasonal cycle parameterization out of the method. Thus, for this reason, the older parts of the age spectra shown here are likely not technically correct, although this primarily applies to the tropical source region age spectra that have a long tail covering many seasonal cycles. In future work, as a means to better compare the measurement-based transport diagnostics to those from model output, we plan to include

the parameterized seasonal cycle in our results.

Another qualification of this method is that it cannot include trace gases with significant production in the atmosphere such as ozone. But we can use the transport diagnostics derived from the other trace gases to help interpret the simultaneously measured ozone in the UTLS. For instance, Figure 15 shows the measured ozone mixing ratios from SEAC⁴RS at the WAS measurement

locations as a function of the modal age. The modal age could be used as an indicator of recent convective injection of air from the surface to the UTLS since a short modal age (days) should correspond with the time of convective injection. We see from Figure 15 that for modal ages of less than a week, essentially all of the ozone mixing ratios are less than 200 ppb with most of them below 100 ppb. For modal ages greater than a week there is a sharp increase in ozone mixing ratios. This is consistent with the expected relationship between recent convective injection and relatively low ozone mixing ratios in the UTLS since the

BL ozone is typically much lower than the background values in this region.

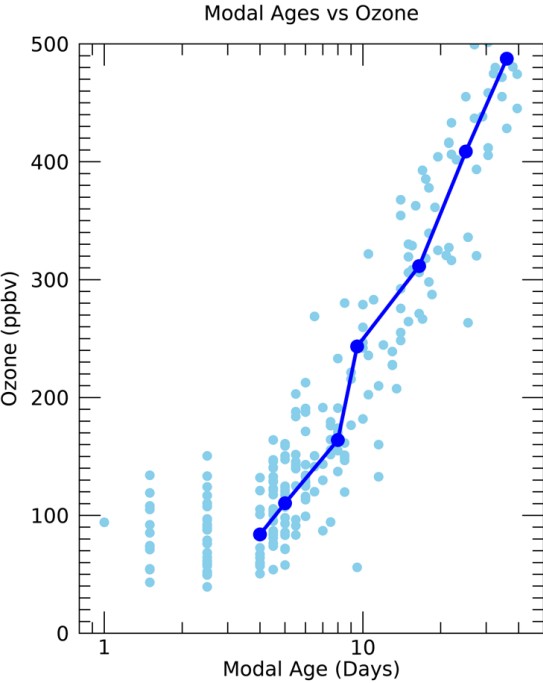

*Figure 15. Ozone mixing ratios vs. modal age for the individual measurements (light blue circles) and theta average profiles (dark blue line and circles) from the SEAC⁴RS mission.*


**Summary**

We have calculated age spectra, path-integrated lifetimes and surface source regions in the UTLS over North America during the summer monsoon season using *in situ* trace gas measurements primarily from the WAS instrument on the ER-2 aircraft during

the SEAC⁴RS campaign. This range of transport diagnostics has not previously been produced solely from measurements for this region of the atmosphere. The results are shown to be broadly consistent with those from previous modeling and measurement studies as well as with our general understanding of large-scale transport and the photochemistry of the trace gases used.

We show that the mean ages in the sampled region range from several months to several years and the modal ages from days to months. The gradients in the ages are primarily a function of theta, but above 450 K there are also substantial latitudinal



gradients such that the oldest air was found at the highest latitudes. Convective injection from the local North American surface was shown to be the most significant source of air to the NAM UTLS below 380 K, with a transition to mostly tropical sources of air in the summer stratospheric overworld. $CO_2$ in particular is useful for identifying surface source regions of air in the UTLS and its use in combination with the wide range of other trace gases is a unique aspect of this study.

The comprehensive utilization of the information deduced from a wide range of simultaneously measured trace gases, following on the methods and ideas from recent studies (e.g. Luo et al., 2018; Hauck et al., 2019, 2020), is an important step forward in our understanding of trace gas distributions in the UTLS and ideally can be done with different data sets in other locations and seasons and lead to improvements in chemistry-climate model transport in the UTLS.

**Acknowledgements**

This work was supported in part by the NOAA Cooperative Agreement with CIRES, NA17OAR4320101. CarbonTracker CT2019B results provided by NOAA ESRL, Boulder, CO, USA from the website at http://carbontracker.noaa.gov. The authors declare no conflicts of interest related to this study.

**Data Availability Statement**

The processed data supporting this study are available from https://csl.noaa.gov/groups/csl8/modeldata.

**Software Availability Statement**

The IDL software used to perform the data analysis and make the figures in this study are available from https://csl.noaa.gov/groups/csl8/modeldata.

**Author contribution**

ER designed and carried out the calculations and wrote the manuscript. EL and SS provided the aircraft measurements. SC, LP, HB and KR provided conceptual support and manuscript suggestions.

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
