# Peer review of "Age Spectra and Other Transport Diagnostics in the North American Monsoon UTLS from SEAC4RS In Situ Trace Gas Measurements"

_Atmospheric Chemistry and Physics, 2021_

## Referee Comment (RC2)

Review of the paper:

"Age Spectra and Other Transport Diagnostics in the North American Monsoon UTLS from SEACRS In Situ Trace Gas Measurements"

written by Ray et al.

**General:**

This is a very interesting and important paper which is worth to be published. It presents a new method how to interpret in situ data observed in the vicinity of the UTLS during one campaign (SEACRS in 2013). By using the age-spectrum technique, the authors quantify: (1) the origin of the sources of the observed species in terms of the boundary layer fractions (tropics or North America), (2) the (path-integrated) lifetimes for the variety of the observed tracer gases, (3) the "outliers", i.e. observations which deviate from a "mean behavior" and are very likely signatures of "polluted" air. In addition, the K-parameter of the age spectrum, assumed here as a 1D spectrum (Hall and Plumb, 1994), was derived from the experimental data and, consequently the age of air of the observed air masses could be calculated. The presented paper can be considered as a pioneering work how to include 20 different species observed at two aircraft during few weeks of a campaign to quantify the transport pathways, timescales of transport, chemical composition of air on its way from the polluted boundary layer to the UTLS region. Although this a is well-written manuscript with almost perfectly designed figures, the formulation of the method is the weakest point of the manuscript and is my major critical point of this review.

**Major point (divided into 3 sub-points):**

- I read your paper 3 times to understand what you are doing. The major difficulty for me was to recognize that you iteratively determine the age spectrum G, starting at the Earth surface and than moving up, level by level. So if you write in line 126 "The surface measurements are convolved with the UTLS age spectra to derive integrated surface boundary conditions" you just killed the reader. You should first introduce equation (1) and (2), boundary conditions, etc, and then introduce your iteration procedure. "To derive integrated surface boundary condition" makes sense for me only within the iteration loop. Maybe a schematic figure would help (like Fig 3 for your assumptions (4) and (5)). Fig 5 is a very good example for the procedure at one level but not for the iteration connecting level $n$ with level $n + 1$.

- I would reserve the word "convolution" only for the equation (1). Equations (7) and (8) are much more "smoother" of the lower boundary and more technical if compared with the main convolution (1)

- The 1d spectrum (Hall and Plumb, 1994) is a very strong simplification. I would state it more clearly at few places like by introducing eq (4) and (5) which are, from my point of view, one of the "smartest ideas" of the paper.

**Minor points:**

- L21: "path-integrated lifetimes" - in my opinion, the partitioning between the NMA and tropical origin is also very important

- L25: "can be compared with chemistry-transport model" - the value of K being a combination of vertical advection and diffusivity is rather difficult to compare...here a better approximation of the 1d age spectrum function would be better but I know that this is not so easy to get an appropriate analytical solution.

- L47: I would not introduce "global lifetimes" which do not play any role for this paper

- general: path-integrated lifetimes - it is not clear what we can win from this concept...better approximation for a "true" lifetime of chemically active species?...would be nice to get more motivation for this concept in the introduction.

- L125: "The surface measurements are convolved with the UTLS age spectra to derive integrated surface boundary conditions for each trace gas"

  ...to derive source latitude distributions (like in Fig 6c), I would concentrate in this first subsection only on the boundary conditions and separate their introduction from the age-spectrum dependent quantities. For the "integrated surface boundary condition" you need something like eq (1). At this stage this equation is not known to the reader, see may major point

- L129: "...so the inclusion of older spectra times does not significantly change the derived surface boundary conditions."

  ...once again, boundary conditions are independent on the age spectrum...however, to calculate age spectrum, boundary conditions in the past (up to 30 years) have to be known

  I think, you mix here two concepts: boundary condition and age spectrum. For this, you need equation (1) that is not present for the reader at this stage of explanation. This is also little bit related to you iteration procedure to derive the age spectrum. I would recommend to introduce these concepts step-by-step: (a) boundary conditions, (b) age spectrum (c) convolution of the boundary condition and age spectrum, i.e. eq.(1) (d) iteration procedure

- L205-210: I would prefer to use $\mu(\tau)$ instead of $\mu - \tau$. Same for $\mu^*$

- L275-280: Eq. (7) and (8): I would not call it "convolution" but much more "smoothing" with a Gaussian smoother. Eqs (9) and (10) should be much more convolutions. The dependence on $y_{TR}$ in (7) and (8) is not present on the right hand side. This part, i.e. eq. (7)-(10), is the weakest part of the description. I think, I know what you want but this is not correctly formulated.

- Eq. (12) and (13) should explain the iteration procedure. I would recommend to do it in more detailed way, using e.g. two levels $n$ and $n + 1$

- L275-305: I understand this part as a formulation of the iteration procedure shown e.g. in Fig 5. but it can certainly be improved (see major point).

- Fig 6, source latitude distributions: this quantity is not well-defined in the paper, please put a reference to the equation in the method description

- Fig 7: please explain how profiles of the path integrated lifetimes were calculated. Is every $\tau_i$ a result of iteration step $i$?

- General: I think that your partition of the source regions into NAM and tropics is a good approximation to demonstrate the method. However, I think the air composition in the tropic during boreal summer is strongly determined by sources within the Asian summer monsoon region. Maybe something for discussion.

---

## Author Comment (AC1)

Responses to reviewer comments on acp-2021-865

Reviewer 1

Minor comment:

I sometimes had a somewhat hard time to follow the reasoning regarding the method. I see that there is a lot of theory behind and it is a balance to not overload the paper with equations and details but still present enough information to enable readers to follow. To make the method available and applicable to more than just a few experts, I suggest to spend some more work on improving the motivation and description of the method details (mainly Sect. 3 and 2). For example, after Eq. 1, the difficulties of applying this relation to measurements could be described, and that certain approximations are indeed necessary. Then, following Eq 3, the approximations used here (which are decribed then in the following) could be summarized.

One problem I had when reading was that it remained sometimes unclear to me what were assumptions to simplify/enable the calculation and what followed from the general theory. A few exemplary related specific comments are given below. As far as I can see, approximations used here include stationarity of age spectrum (e.g., no seasonality), the inverse Gaussian shaped 1D spectrum, usage of altitude z instead of tracer-based equivalent height, restriction to / choice of 2 surface source regions. It is clear to me that such approximations need to be done to make the method applicable to real data. I'd only suggest to clarify at several places what results from general theory (as e.g. provided by Holzer and Hall, 2000) and what are additional assumptions/approximations. As said above, a summary of these in Sect. 3 and maybe also an enhanced discussion (e.g., L720ff) could help the reader.

Specific comments:

L24: "... provide a range of transport diagnostics ...". I find this formulation somehow vague and suggest to be more specific - e.g. state which diagnostics explicitly.

Added the specific transport diagnostics referred to in this sentence.

L107: Perhaps better say "CO2 surface mixing ratios ..."

Added 'boundary layer' here since the bottom three levels in CarbonTracker were used as described in the supplement (Section S1).

L126: As far as I can see, figure 3 is the first figure mentioned in the text. Perhaps change the order of figures - if this makes sense...

You are correct that this is the first figure mentioned and that it doesn't necessarily make sense that way. Rather than switch figure placement the reference to Figure 3 has been removed here since it's not really necessary. The references in the sentence are meant to support the range of surface latitude influence to the UTLS over North America while Figure 3 is simply a schematic representation.

L194: If I understand the theory correctly, including time-dependence into the age spectrum via letting K be a function of t is not equivalent to considering a time-depending transport operator in the continuity equation. (The specific form of G in Eq. (2) is the Greens function only for stationary 1D diffusive transport). If I'm correct, I suggest to clarify this here and say that this is another assumption.

This is a good point. The time dependence of K has been removed in Eq. 2 since we really only consider a short time during the SEAC4RS mission during which period the variation of the age spectra will be much more dependent on location compared to time over the course of the mission.

L200ff: Isn't this compact relationship equivalent to the relationship between scale height and lifetime found by Ehhalt et al. (2007, their Fig. 7)?

You are correct, this is essentially the same relationship. A reference to Ehhalt et al., 2007 has been added here.

L212: "... do not assume steady state conditions ..." Is this true in general? I understand that the surface mixing ratio is allowed to depend on time, but that for the age spectrum still steady state conditions are assumed. (See also my comment above regarding L194). Please clarify.

You are correct, we do actually assume steady state conditions in the calculation so this has been removed.

L251: But this assumption neglects the pathway described in L230, that air from NA surface can be transported into the lower stratosphere via the tropical upper troposphere. Is there a sound reasoning why this pathway can be neglected, or is it just to make the computation feasible? In the same spirit, are there good reasons why transport from other extratropical surface regions beside North America to the sampling region can be neglected?

We actually don't neglect this pathway since we consider transport from the NA surface up to time scales of 200 days as discussed in the paragraph following Eqs. 4 and 5 and the value of  $t_f$ . The sentences around line 251 state that we assume the shortest time scales are primarily from the NA surface and we added that the longest time scales are primarily from the tropics. We also added a sentence that the time scales on the order of months can have significant contributions from both regions.

The main reason we don't explicitly consider transport from other extratropical source regions is that we don't have surface measurements from many other locations, especially Asia. As mentioned by Reviewer 2, the Asian monsoon region is a significant source of air to the extratropical UTLS during boreal summer so there is likely to be influence from this region in the sampled region. We add some discussion of this possibility in the paragraph containing line 230 and also in the Discussion section.

L253: Related to the above comment, I'd suggest to write here something like "We assume that the age spectra can be partitioned as ...".

Done

Eqns. 4/5: I don't understand the separation here into f and G functions. First, the partitioning into scaling factors and G I see as an empirical ansatz - is this correct? (If yes, I'd suggest to state that). And why are transport parameters from tropical and NH surface patches the same? Shouldn't the G's on the rhs of the equations actually be age spectra for the specific surface patches (e.g. as in Hauck et al., 2020, Eq. 12)? This would then also affect the rearrangement leading to Eq. 12. Maybe a few more words for explanation could be helpful, in the sense whether this ansatz follows from the general theory or is an assumption.

This paragraph has been modified to try to address these comments and to be more consistent with the formulation used in the Hauck et al., 2020 study. We have included a new Eq. 4 that is essentially the same as Eq. 6 in Hauck et al. and moved up and modified Eq. 6, now Eq. 5, that is now similar to Eq. 7 in Hauck et al.

Eqns. 7/8: I don't fully get the meaning of these equations here? E.g., why is there a  $y_TR$  dependence on the lhs if the latitude dependence is integrated out? Or should this be  $y_pTR$  as on the rhs?

Yes, the dependence on the left-hand side should be y\_pTR so this has been changed. These equations are meant to show

L275ff: Also here, regarding the steps leading to Eq. 11, it is not clear to me whether these follow from theory, or are further approximations to construct a surface mixing ratio time series indepedent of transit time which can be pulled out of the integral.

Additional description has been added here to make it clear what follows from theory and previous studies and what assumptions and approximations are new to this study.

L480-L515: I find the description of the various scalings applied rather complicated, technical and confusing. Perhaps, these paragraphs could be moved to an appendix and just very briefly summarized here, to not distract the reader's focus?

This section of text and description of the scalings has been moved to the supplement section S5.

L645: I think two other reasons which likely contribute to the differences between the here observed and recently published surface source region fractions are: 1) SEAC4RS focussed on sampling strong convection where the local source impact is likely much stronger than in the zonal mean; 2) The models used in previous studies likely underestimate the convective impact.

Agreed, these are good reasons for the differences. We do mention the convectively active NAM region in the previous paragraph. We have added the likely underestimation of convective impact on the UTLS in the paragraph you reference.

L737: "... primarily applies to the tropical source region age spectra ..." I'm not sure whether this can be stated. The studies by Hauck et al. (2020) and Yan et al. (2021) showed also very clear seasonality for extratropical source region age spectra, which in relative terms could be even clearer than for tropical spectra. If I'm correct, I'd suggest to just delete this part of the sentence.

**The end of the sentence has been deleted since you are likely correct.**

L768: "... most significant source of air ..." In view of the presented results (e.g., Fig. 10c) I'd rather say "... a significant source ...". The NA fraction is indeed substantial, but on average below 50% (as far as I can see).

**Agreed, changed.**

Figure S1: What is the meaning of the intensity/darkness of orange shading? Would be good to specify that in the caption.

The shading represents theta level and this has been added to the caption.

Technical corrections: L144: Isn't "extrapolated" instead of "interpolated" what you mean here?

**Yes, changed.**

L466: Isn't it "Eq. 3" which relates mu to G and should be referred to here?

Yes, changed.

Reviewer 2

Major point (divided into 3 sub-points):

• I read your paper 3 times to understand what you are doing. The major difficulty for me was to recognize that you iteratively determine the age spectrum G, starting at the Earth surface and than moving up, level by level. So if you write in line 126 "The surface measurements are convolved with the UTLS age spectra to derive integrated surface boundary conditions" you just killed the reader. You should first introduce equation (1) and (2), boundary condi- tions, etc, and then introduce your iteration procedure. "To derive integrated surface boundary condition for measurements are convolved with the under the procedure of the procedure of the procedure integrated surface boundary conditions" you just killed the reader. You should first introduce equation (1) and (2), boundary condi- tions, etc, and then introduce your iteration procedure. "To derive integrated surface boundary condition" makes sense for me only within the iteration loop. Maybe a schematic figure would help (like Fig 3 for your assumptions (4) and (5)). Fig 5 is a very good example for the procedure at one level but not for the iteration connecting level n with level n + 1.

Much of the methods section has been rewritten to try to make it more clear and to address this and other reviewer comments.

• I would reserve the word "convolution" only for the equation (1). Equations (7) and (8) are much more "smoother" of the lower boundary and more technical if compared with the main convolution (1)

Done

• The 1d spectrum (Hall and Plumb, 1994) is a very strong simplification. I would state it more clearly at few places like by introducing eq (4) and (5) which are, from my point of view, one of the "smartest ideas" of the paper.

We include some additional discussion of the age spectra shape following Equations 4 and 5. In the initial stages of this research we did not specify the shape of G but could not justify that the features found in the optimized G were unique. Specifying the shape of G allowed us to more easily solve for the additional transport diagnostics so it was a tradeoff, but a necessary one in our view.

Minor points:

• L21: "path-integrated lifetimes" - in my opinion, the partitioning between the NMA and tropical origin is also very important

This sentence was modified from 'and surface source regions' to 'and partitioning between North American and tropical surface source origins' to emphasize your point that this is one of the key results.

• L25: "can be compared with chemistry-transport model" - the value of K being a combination of vertical advection and diffusivity is rather difficult to compare...here a better approximation of the 1d age spectrum function would be better but I know that this is not so easy to get an appropriate analytical solution.

Yes, the original version of the calculation had no prescribed form of the age spectra and the results were interesting but hard to justify the individual features. The tradeoff in prescribing the age spectra shape with a value of K is that it provides a reasonable comparison with previously estimated age spectra and ideally more confidence in the other diagnostics such as the path-integrated lifetimes. This sentence in the abstract is really referring to the comparison of the age spectra and other transport diagnostics with those from model output and this work has already begun.

• L47: I would not introduce "global lifetimes" which do not play any role for this paper

**Removed**

• general: path-integrated lifetimes - it is not clear what we can win from this concept...better approximation for a "true" lifetime of chemically active species?...would be nice to get more motivation for this concept in the introduction.

The path-integrated lifetime is foremost a necessity for the calculation performed here but also your question about its usefulness would perhaps be a common one among readers since it is a rarely used quantity. A sentence has been added in this paragraph to add some motivation for its usefulness but the essential point is that it is a required part of the calculation and it is technically a transport diagnostic since it reveals broad path information.

• L125: "The surface measurements are convolved with the UTLS age spectra to derive integrated surface boundary conditions for each trace gas"

...to derive source latitude distributions (like in Fig 6c), I would concentrate in this first subsection only on the boundary conditions and separate their introduction from the age- spectrum dependent quantities. For the "integrated surface boundary condition" you need something like eq (1). At this stage this equation is not known to the reader, see may major point

Thank you for this comment! The description of the method is rather extended with many details so any suggestions on how to make it easier to read and understand are appreciated. A version of this sentence has been moved down into Section 3 in the paragraph discussing the boundary conditions following Equation 3.

• L129: "...so the inclusion of older spectra times does not significantly change the derived surface boundary conditions."

...once again, boundary conditions are independent on the age spectrum...however, to calculate age spectrum, boundary conditions in the past (up to 30 years) have to be known

I think, you mix here two concepts: boundary condition and age spectrum. For this, you need equation (1) that is not present for the reader at this stage of explanation. This is also little bit related to you iteration procedure to derive the age spectrum. I would recommend to introduce these concepts step-by-step: (a) boundary conditions, (b) age spectrum (c) convolution of the boundary condition and age spectrum, i.e. eq.(1) (d) iteration procedure

Again, thank you for this comment. This sentence has also been removed from this section and a version of it added in Section 3.

• L205-210: I would prefer to use  $\mu(\tau)$  instead of  $\mu - \tau$ . Same for  $\mu^*$

**Done**

• L275-280: Eq. (7) and (8): I would not call it "convolution" but much more "smoothing" with a Gaussian smoother. Eqs (9) and (10) should be much more convolutions. The dependence on  $y_{TR}$  in (7) and (8) is not present on the right hand side. This part, i.e. eq. (7)-(10), is the weakest part of the description. I think, I know what you want but this is not correctly formulated.

These are good comments and further description has been added to this section also in response to the comments of Reviewer 1. As suggested, the use of

convolution has been removed when describing Eqs. 7 and 8 (now 6 and 7) and the dependence on the left hand side is now y\_pTR and y\_pNA to represent dependence on the peaks of the latitudinal distributions L.

• Eq. (12) and (13) should explain the iteration procedure. I would recommend to do it in more detailed way, using e.g. two levels n and n + 1

Thank you for the suggestion. Much of this section has been rewritten and a list of steps describing the method has been included to hopefully make it more clear.

• L275-305: I understand this part as a formulation of the iteration procedure shown e.g. in Fig 5. but it can certainly be improved (see major point).

Thank you again, as mentioned above much of this section has been rewritten to address this and other comments.

• Fig 6, source latitude distributions: this quantity is not well-defined in the paper, please put a reference to the equation in the method description

A reference to the latitudinal distributions (L) is now included in the caption. These distributions are included in Equations 8 and 9 and shown in Figure S4.

• Fig 7: please explain how profiles of the path integrated lifetimes were calculated. Is every  $\tau_i$  a result of iteration step i?

The calculation of the path integrated lifetimes is described in the Methods section and in Lines 406-411 of the original manuscript just before Figure 7 appears in the paper. The subscript i refers to the trace gas i as originally described in Equation 1 and referred to in all subsequent equations.

• General: I think that your partition of the source regions into NAM and tropics is a good approximation to demonstrate the method. However, I think the air composition in the tropic during boreal summer is strongly determined by sources within the Asian summer monsoon region. Maybe something for discussion.

You are correct and this is a good point worth including. We added some discussion of the Asian monsoon in the Methods section around line 230 and in the Discussion section. It would be nice if we had sufficient surface measurement time series in the Asian monsoon region so that we could include it as a source region in the calculation.

---

## Referee Report (RR1)

Review of the revised version of the paper:

"Age Spectra and Other Transport Diagnostics in the North American Monsoon UTLS from SEACRS In Situ Trace Gas Measurements"

written by Ray et al.

**General:**

I like the revised version very much. I have only few optional minor points

**Minor points:**

- L185
  I would replace wording "concentrations" by "mixing ratios". This is what you need in the following

- L360
  After point 5 I would expect something like: By varying K and the source coordinates ypTR and ypNA, the best fit of the tropical and NA spectra will be determined.

---

## Author Response (AR2)

Responses to reviewer comments

We thank the referee for the positive comments.  The two specific comments are addressed below.

- L185
  I would replace wording "concentrations" by "mixing ratios". This is what you need in the following

Done

- L360
  After point 5 I would expect something like: By varying K and the source coordinates ypTR and ypNA, the best fit of the tropical and NA spectra will be determined.

The determination of K, ypTR and ypNA are actually found in the optimization listed as Step 4.  We have added those specific terms in that step following the phrase 'transport parameters'.